# Coordination of peptidoglycan synthesis and outer membrane constriction during *Escherichia coli* cell division

Andrew N Gray[1†], Alexander JF Egan[2†], Inge L van't Veer[3], Jolanda Verheul[4], Alexandre Colavin[5], Alexandra Koumoutsi[6], Jacob Biboy[2], A F Maarten Altelaar[7], Mirjam J Damen[7], Kerwyn Casey Huang[5,8,9], Jean-Pierre Simorre[10], Eefjan Breukink[3], Tanneke den Blaauwen[4]*, Athanasios Typas[6]*, Carol A Gross[1,11,12]*, Waldemar Vollmer[2]*

[1]Department of Microbiology and Immunology, University of California, San Francisco, San Francisco, United States; [2]Centre for Bacterial Cell Biology, Institute for Cell and Molecular Biosciences, Newcastle University, Newcastle Upon Tyne, United Kingdom; [3]Membrane Biochemistry and Biophysics, Bijvoet Centre for Biomolecular Research, University of Utrecht, Utrecht, The Netherlands; [4]Bacterial Cell Biology, Swammerdam Institute for Life Sciences, Faculty of Science, University of Amsterdam, Amsterdam, The Netherlands; [5]Biophysics Program, Stanford University, Stanford, United States; [6]Genome Biology Unit, European Molecular Biology Laboratory Heidelberg, Heidelberg, Germany; [7]Biomolecular Mass Spectrometry and Proteomics, Bijvoet Centre for Biomolecular Research and Utrecht Institute for Pharmaceutical Sciences, University of Utrecht, Utrecht, The Netherlands; [8]Department of Bioengineering, Stanford University, Stanford, United States; [9]Department of Microbiology and Immunology, Stanford University School of Medicine, Stanford, United States; [10]Institut de Biologie Structurale, Université Grenoble Alpes, Grenoble, France; [11]Department of Cell and Tissue Biology, University of California, San Francisco, San Francisco, United States; [12]California Institute of Quantitative Biology, University of California, San Francisco, San Francisco, United States

*For correspondence:
t.denblaauwen@uva.nl (TdB); typas@embl.de (AT); cgrossucsf@gmail.com (CAG); w.vollmer@ncl.ac.uk (WV)

†These authors contributed equally to this work

Competing interests: The other authors declare that no competing interests exist.

**Abstract** To maintain cellular structure and integrity during division, Gram-negative bacteria must carefully coordinate constriction of a tripartite cell envelope of inner membrane, peptidoglycan (PG), and outer membrane (OM). It has remained enigmatic how this is accomplished. Here, we show that envelope machines facilitating septal PG synthesis (PBP1B-LpoB complex) and OM constriction (Tol system) are physically and functionally coordinated via YbgF, renamed CpoB (**C**oordinator of **P**G synthesis and **O**M constriction, associated with PBP1**B**). CpoB localizes to the septum concurrent with PBP1B-LpoB and Tol at the onset of constriction, interacts with both complexes, and regulates PBP1B activity in response to Tol energy state. This coordination links PG synthesis with OM invagination and imparts a unique mode of bifunctional PG synthase regulation by selectively modulating PBP1B cross-linking activity. Coordination of the PBP1B and Tol machines by CpoB contributes to effective PBP1B function in vivo and maintenance of cell envelope integrity during division.

## Introduction

Cell shape and osmotic stability are maintained by the stress-bearing peptidoglycan (PG) sacculus (cell wall) in nearly all bacteria (*Vollmer et al., 2008*). The sacculus, a continuous mesh-like structure of glycan

**eLife digest** All bacterial cells are surrounded by a membrane, which forms a protective barrier around the cell. Most bacteria also have a wall surrounding the membrane, which provides structural support. When a bacterial cell divides to produce two daughter cells, it produces a belt-like structure around the middle of the cell. This brings the membrane and cell wall on each side together to a 'pinch-point' until the two halves of the cell have been separated. This process must be carefully controlled to ensure that the cell does not burst open at any point.

Some bacteria known as 'Gram-negative' bacteria have a second membrane on the other side of the cell wall. These cells divide in the same way as other bacteria, but the need to coordinate the movement of three structures instead of two makes it more complicated. Many proteins are known to be involved. For example, one group (or 'complex') of proteins—which includes a protein called PBP1B—helps to produce new cell wall material. Another complex called the Tol system provides the energy needed for the outer membrane to be pulled inwards towards the pinch point. However, it has not been clear how these complexes work together to allow the cell to divide.

Here, Gray, Egan et al. searched for proteins that can interact with PBP1B during cell division in the Gram-negative bacterium *E. coli*. The experiments found that a protein called CpoB interacts with both PBP1B and the Tol system. CpoB is found in a band around the middle of the cell, and it regulates the activity of PBP1B in response to signals from the Tol system. If the activity of CpoB is disrupted, cell wall production and the movement of the outer membrane are no longer coordinated, and the membrane falls apart, leading to the death of the bacteria.

Gray, Egan et al.'s findings show how the production of new cell wall material can be linked to the inwards movement of the outer membrane during cell division. The next challenges are to understand the precise details of how these processes are coordinated by CpoB and to find out whether CpoB also plays the same role in other bacteria.

strands cross-linked by short peptides, encases the inner (cytoplasmic) membrane (IM) and is essential for viability. Several prominent classes of antibiotics (e.g., β-lactams and glycopeptides) inhibit PG synthesis, causing lysis and cell death (*Schneider and Sahl, 2010*). In Gram-negative bacteria, the outer membrane (OM), an asymmetric bilayer of phospholipids and lipopolysaccharides, surrounds the mostly single-layered sacculus (*Gan et al., 2008*) and forms a vital permeability barrier (*Nikaido, 2003*). Covalent and non-covalent interactions between abundant OM proteins (Lpp, Pal, OmpA) and PG tether the OM to the sacculus, maintaining OM stability (*Hantke and Braun, 1973*; *Parsons et al., 2006*). Together, the IM, PG, and OM comprise the tripartite Gram-negative cell envelope (*Figure 1A*). Bacteria must synchronize growth and division of these layers, as imbalanced growth could lead to breaches that compromise the permeability barrier or even the structural integrity of the cell. However, we do not yet understand the mechanisms that accomplish this synchronization.

Coordinating growth across the layers of the Gram-negative bacterial cell envelope is complex, particularly since all energy and precursors for assembling and constricting these layers must come from the cytoplasm. To overcome this challenge, bacteria utilize IM-associated multicomponent machineries that span the entire envelope. Two machineries, organized by distinct cytoskeletal elements, assemble and disassemble in a cell-cycle-regulated manner and mediate different phases of sacculus growth (*Typas et al., 2012*): (1) the cell elongation machinery (elongasome), organized by the actin homolog MreB, mediates lateral PG synthesis along the length of the cell, and (2) the cell division machinery (divisome), organized by the tubulin homolog FtsZ, mediates new pole synthesis at the septum (*Egan and Vollmer, 2013*). These complex machineries are comprised of structural and regulatory subunits, components with distinct functions (e.g., DNA segregation, PG precursor synthesis and transport), and PG biosynthetic and modifying enzymes.

Sacculus growth is orchestrated by a repertoire of PG synthases, including glycosyltransferases (GTases) that polymerize glycan strands from the precursor saccharide moiety lipid II, transpeptidases (TPases) that cross-link peptides between adjacent glycan strands, and bifunctional PG synthases that carry out both activities (*Typas et al., 2012*). The two *Escherichia coli* monofunctional TPases, PBP2, and PBP3, are essential subunits of the elongasome and the divisome, respectively. Likewise, the two major bifunctional PG synthases, PBP1A and PBP1B, participate predominantly in elongation and

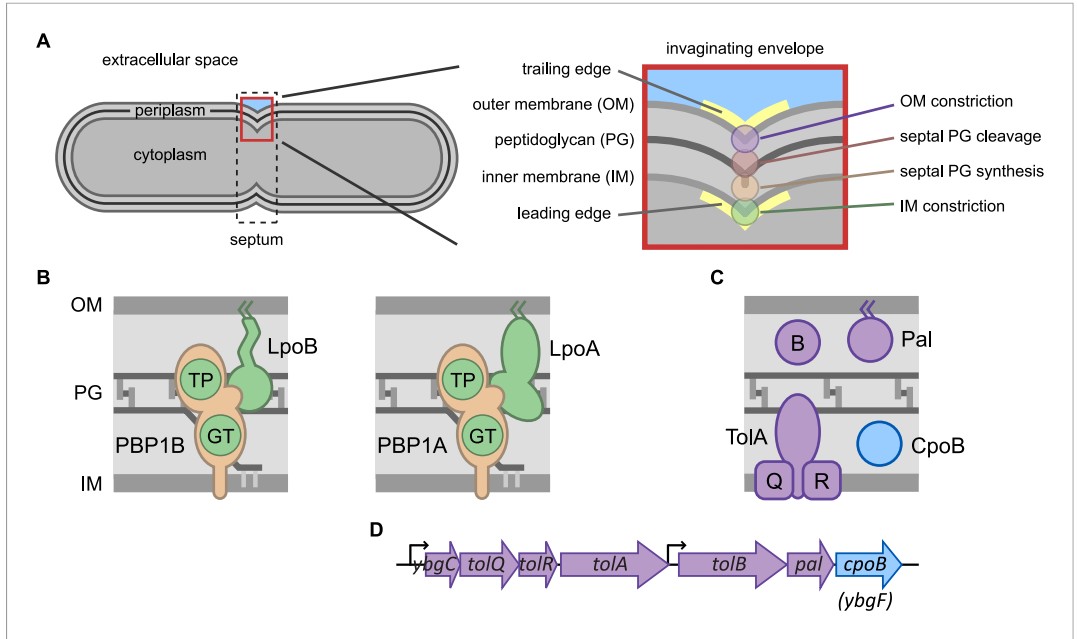

**Figure 1**. Envelope constriction in Gram-negative bacteria and related protein machines. (**A**) Illustration of the tripartite Gram-negative cell envelope. In zoom (right), processes involved in envelope constriction. (**B**) Major bifunctional PG synthases of *E. coli*, capable of both glycan strand elongation (glycosyltransferase activity, GT) and peptide cross-linking (transpeptidase activity, TP). The IM-localized synthases are activated by cognate regulatory OM lipoproteins. (**C**) Proteins encoded by the *tol-pal* operons (**D**).

division, respectively (*Bertsche et al., 2006*; *Typas et al., 2010*; *Banzhaf et al., 2012*). However, in contrast to the monofunctional TPases, which are dedicated to their respective roles, the bifunctional synthases can partially substitute for each other, enabling cells to survive with only one of them (*Yousif et al., 1985*). These IM-localized bifunctional synthases have obligate cognate regulatory OM lipoproteins, LpoA and LpoB, which are required for activity in vivo (*Paradis-Bleau et al., 2010*; *Typas et al., 2010*). The Lpo activators span most of the periplasm (∼210 Å in width; *Matias et al., 2003*) and traverse the sacculus (∼40–60 Å pore size; *Demchick and Koch, 1996*; *Vazquez-Laslop et al., 2001*) to interact with their partner PBPs (*Egan et al., 2014*; *Jean et al., 2014*), forming trans-envelope PG synthase complexes (*Figure 1B*).

Electron microscopy studies first indicated that distances between the OM, PG, and IM remain remarkably consistent throughout cell division, providing an early indication that envelope constriction processes occur in close proximity to each other and are tightly coordinated (*Weigand et al., 1976*; *Fung et al., 1978*; *MacAlister et al., 1987*; *Bi and Lutkenhaus, 1991*). It is now clear that IM constriction, PG synthesis, and subsequent PG hydrolysis to separate daughter cells (septal cleavage) are coordinated via the divisome. FtsZ forms a ring-like structure in the cytoplasm that provides the membrane contractile force (*Osawa et al., 2009*), and together with FtsA (*Szwedziak et al., 2012*; *Osawa and Erickson, 2013*; *Loose and Mitchison, 2014*; *Szwedziak et al., 2014*) serves as a scaffold for divisome assembly, including recruitment of PG synthases and hydrolases (*Egan and Vollmer, 2013*). Septal PG synthesis, principally orchestrated by PBP3 and PBP1B (*Bertsche et al., 2006*), occurs at the leading edge of the inward-moving septum, adjacent to the invaginating IM (*Figure 1A*). Septal cleavage, controlled by tightly regulated periplasmic amidases (*Heidrich et al., 2001*; *Uehara et al., 2010*), follows closely after synthesis and adjacent to the invaginating OM. Both topological constraints and regulatory input from IM and/or OM proteins ensure tight spatial regulation of septal cleavage (*Uehara et al., 2010*; *Yang et al., 2011*).

OM constriction is promoted by the energy-transducing Tol system, which localizes to mid-cell during the later stages of cell division in a divisome-dependent manner (*Gerding et al., 2007*). IM proteins TolQ, TolR, and TolA, which form a complex (*Derouiche et al., 1995*; *Lazzaroni et al., 1995*; *Journet et al., 1999*), periplasmic TolB, and OM lipoprotein Pal are all encoded in two adjacent

operons (*Figure 1C,D*). Loss of any of these components results in delayed OM constriction and defects in OM integrity, leading to OM blebbing, periplasmic leakage, and pleiotropic drug and stress sensitivities (*Bernadac et al., 1998*; *Cascales et al., 2002*; *Gerding et al., 2007*). For function, Tol harnesses proton motive force (PMF) via TolQR, a homolog of the flagellar motor MotAB (*Cascales et al., 2001*). This has been proposed to energize TolA, inducing it to adopt an extended conformation and interact with TolB and/or Pal (*Cascales et al., 2000*; *Germon et al., 2001*; *Lloubes et al., 2001*); cycles of Tol–Pal interaction and release are then thought to promote OM invagination (*Gerding et al., 2007*). This interaction model has been challenged, however (*Bonsor et al., 2009*), and the mechanism by which the Tol system promotes OM constriction remains to be fully elucidated. Further, how Tol-facilitated OM constriction is coordinated with septal PG synthesis and other envelope constriction processes has remained completely unknown.

We previously identified a genetic link between PBP1B-LpoB and Tol (*Typas et al., 2010*). Here, we report that physical and functional coordination of the two machines is required to properly synchronize PG synthesis and OM constriction during cell division. We implicate YbgF, previously of unknown function, in mediating this coordination, and therefore name it CpoB, or Coordinator of PG synthesis and OM constriction, associated with PBP1B. We show that CpoB, PBP1B-LpoB, and Tol localize concurrently to the septum during cell division, and interact to form a higher-order complex that spatially links PG synthesis and OM invagination. These physical interactions are dynamic and allow direct regulation of PBP1B activity in response to Tol assembly and cycles of PMF utilization. CpoB is required for proper PBP1B function in vivo, and loss of CpoB-mediated coordination between PBP1B and Tol leads to defects in OM integrity, illustrating the importance of mechanisms that ensure coordination of cell division processes across the envelope.

## Results

### The auxiliary Tol-associated protein CpoB has a PBP1B-associated function

To identify novel regulators of PG synthesis during cell division, we queried the *E. coli* chemical genomics database (*Nichols et al., 2011*) for deletion strains whose growth responses closely resembled those of a strain lacking PBP1B (encoded by *mrcB*) across a range of drug and environmental stress conditions. This approach previously identified LpoB, the OM lipoprotein activator of PBP1B (*Typas et al., 2010*). Δ*cpoB* exhibited the second highest correlation with Δ*mrcB* (*Figure 2A*; correlation coefficient = 0.47, $p < 10^{-18}$), suggesting that CpoB may also be functionally associated with PBP1B. CpoB (YbgF) is encoded by the last gene in the *tol-pal* operon and binds TolA in vitro (*Krachler et al., 2010*), but its deletion does not cause the severe OM integrity defects typically associated with *tol-pal* deletions (*Vianney et al., 1996*). Thus, CpoB is not critical for Tol function, and its cellular role has remained enigmatic.

Δ*cpoB* and Δ*mrcB* mutants shared increased sensitivities to multiple β-lactams (*Figure 2A*), including cefsulodin, which preferentially inhibits PBP1A. As loss of both PBP1A and PBP1B function is synthetically lethal (*Yousif et al., 1985*), the enhanced cefsulodin sensitivity of Δ*cpoB* likely derives from defects in PBP1B function. Consistent with this idea, overexpression of PBP1B abrogated the cefsulodin sensitivity of Δ*cpoB* (*Figure 2B*). Additionally, both Δ*cpoB* and Δ*mrcB* mutants exhibited increased lysis in an envelope stability screen (*Paradis-Bleau et al., 2014*). Importantly, the lysis phenotype of Δ*cpoB* Δ*mrcB* did not exceed that of the Δ*mrcB* single mutant (*Figure 2—figure supplement 1A,B*). This epistatic relationship strongly suggests that CpoB acts via PBP1B. Expression of CpoB in trans complemented Δ*cpoB* phenotypes, confirming that they result from loss of CpoB rather than indirect effects on upstream *tol-pal* gene expression (*Figure 2—figure supplement 2*).

Having established a functional relationship between CpoB and PBP1B, we asked whether they interact physically in vivo. Indeed, we identified CpoB in a screen for novel PBP1B interaction partners, using in vivo photo-cross-linking following incorporation of a non-natural amino acid (*p*Bpa) at specific exposed sites (*Chin and Schultz, 2002*). Initially focusing on the PBP1B UB2H domain, we identified two sites (T118 and E123), of 19 examined, that cross-linked to a protein of approximately 30 kDa (*Figure 3A,B*). These sites reside in a cleft between the UB2H and TPase domains. Testing opposing amino acids in the TPase domain (T751 and T753) yielded cross-link products of the same size (*Figure 3A,B*). Mass spectrometric analysis of the cross-link bands identified CpoB as a main constituent of the cross-linking product (*Table 1*). These results imply that PBP1B and CpoB physically interact in vivo.

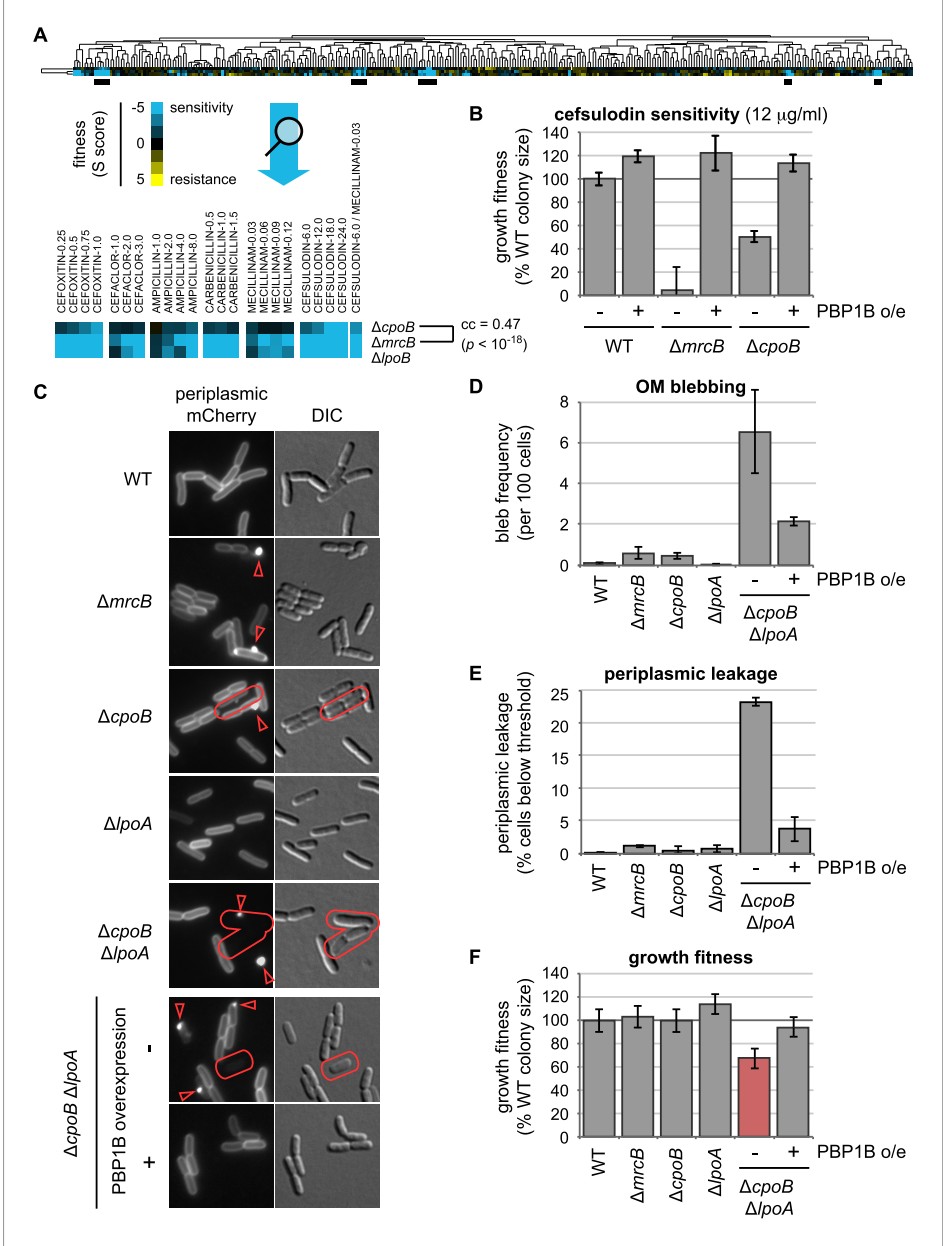

**Figure 2**. Shared chemical sensitivities and genetic interactions implicate CpoB in PBP1B function. (**A**) Chemical genetic phenotype profiles for strains lacking PBP1B (encoded by *mrcB*), LpoB, and CpoB, from hierarchical clustering of the growth fitnesses of 3979 gene deletions across 324 chemical stress conditions (*Nichols et al., 2011*). Blue, low fitness (sensitivity); yellow, high fitness (resistance). Δ*mrcB* and Δ*cpoB* (Δ*ybgF*) showed significant similarity. Underlined regions correspond to strongest shared sensitivities and are magnified below the full profile. (**B**) Sensitivity of Δ*cpoB* to cefsulodin, predominantly an inhibitor of PBP1A, is alleviated by PBP1B overexpression. Growth fitness was assessed by pinning cells to agar plates in a 1536-spot format and quantifying colony size after 12–14 hr of growth. Error bars depict standard deviations ($n \geq 44$). (**C**) CpoB-associated envelope defects visualized by fluorescence microscopy using periplasmic mCherry. Arrowheads, fluorescent foci indicating OM blebs or vesicles; outlines, loss of peripheral fluorescence due to periplasmic leakage. (**D**) and (**E**). Quantification of microscopy phenotypes: (**D**) OM blebbing, and (**E**) loss of periplasmic fluorescence, indicative of periplasmic leakage. Threshold value was established based on quantification of cells with known severe OM defects (Δ*tolA*). Cells lacking CpoB exhibited envelope defects that were severely exacerbated in the absence of LpoA and alleviated by PBP1B overexpression. (**F**) Quantification of growth

*Figure 2. continued on next page*

*Figure 2. Continued*

fitness for indicated mutant strains. A strain lacking CpoB and LpoA exhibited a synthetic growth defect that was alleviated by PBP1B overexpression.
The following figure supplements are available for figure 2:

**Figure supplement 1**. Elevated cell lysis in cells lacking PBP1B or CpoB; loss of LpoA exacerbates lysis in the absence of CpoB.

**Figure supplement 2**. Complementation of Δ*cpoB* phenotypes.

**Figure supplement 3**. LpoA encodes a second function, and the LpoA TPR domain is dispensable for PBP1A activation.

**Figure supplement 4**. The N-terminal TPR domain of LpoA encodes a novel function that compensates for loss of CpoB.

## Genetic interactions further implicate CpoB in PBP1B function and reveal a CpoB-related function for LpoA

If CpoB contributes to PBP1B function, then absence of PBP1A or its required activator LpoA (*Paradis-Bleau et al., 2010*; *Typas et al., 2010*) should result in an aggravated phenotype (*Figure 2—figure supplement 3C*). Consistent with this expectation, a strain lacking both CpoB and LpoA showed a severe growth defect (*Figure 2F*). To our surprise, however, a strain lacking both CpoB and PBP1A grew normally (*Figure 2—figure supplement 3A*). We tracked down the reason for this discrepancy to a previously unknown function of LpoA. LpoA possesses two-domains, an N-terminal tetratricopeptide repeat (TPR) domain and a C-terminal domain (*Figure 2—figure supplement 4A*). The C-terminal domain is sufficient to bind and activate PBP1A in vitro (*Typas et al., 2010*; *Jean et al., 2014*) and, we show here, in vivo (*Figure 2—figure supplement 3B*). We found that the N-terminal TPR domain encodes an additional function that can compensate for loss of CpoB. As evidence, though a Δ*cpoB* Δ*mrcA* mutant has no growth defect, a triple mutant also lacking the LpoA TPR domain, Δ*cpoB* Δ*mrcA lpoA*(Δ*TPR*), recapitulates the growth defect of the Δ*cpoB* Δ*lpoA* mutant (*Figure 2—figure supplement 4D*). Importantly, this CpoB compensatory function of LpoA is required only when PBP1B is the sole active bifunctional synthase, as a Δ*cpoB lpoA*(Δ*TPR*) mutant (in which PBP1A is active) grows normally (*Figure 2—figure supplement 4D*).

To further assess the importance of CpoB function for envelope integrity, we used periplasmic mCherry to fluorescently visualize envelope morphology (*Gerding et al., 2007*). Both Δ*mrcB* and Δ*cpoB* strains exhibited minor OM defects, including elevated levels of OM blebbing (*Figure 2C,E*). As with the growth phenotype, these Δ*cpoB* envelope defects were greatly exacerbated in the absence of LpoA, with extensive OM blebbing, periplasmic leakage and cell lysis (*Figure 2C,E* and *Figure 2—figure supplement 1A–C*). All of these effects were recapitulated in a Δ*cpoB* Δ*mrcA lpoA* (Δ*TPR*) triple mutant (*Figure 2—figure supplement 4B–D*), and ameliorated by overexpression of PBP1B (*Figure 2C–F* and *Figure 2—figure supplement 4B–D*). Of note, the envelope defects of Δ*cpoB* Δ*lpoA* were distinct from those of *tol-pal* mutants (*Gerding et al., 2007*), with more extensive lysis but less OM blebbing and periplasmic leakage (*Figure 2C–E* and *Figure 2—figure supplement 1C*). Taken together, these results validate further that CpoB contributes to proper PBP1B function in vivo; indicate that this function is important for envelope integrity, and particularly important when cells must rely on PBP1B; and serendipitously reveal a partial redundancy of function between CpoB and the LpoA TPR domain.

## CpoB localizes to the septum late in cell division, coincident with the onset of OM constriction

We next investigated whether CpoB, like PBP1B and Tol (*Bertsche et al., 2006*; *Gerding et al., 2007*), is recruited to the septum. Using a functional, endogenously expressed CpoB-mCherry fusion protein (*Figure 4—figure supplement 1A*), we observed that CpoB localized at sites of visible

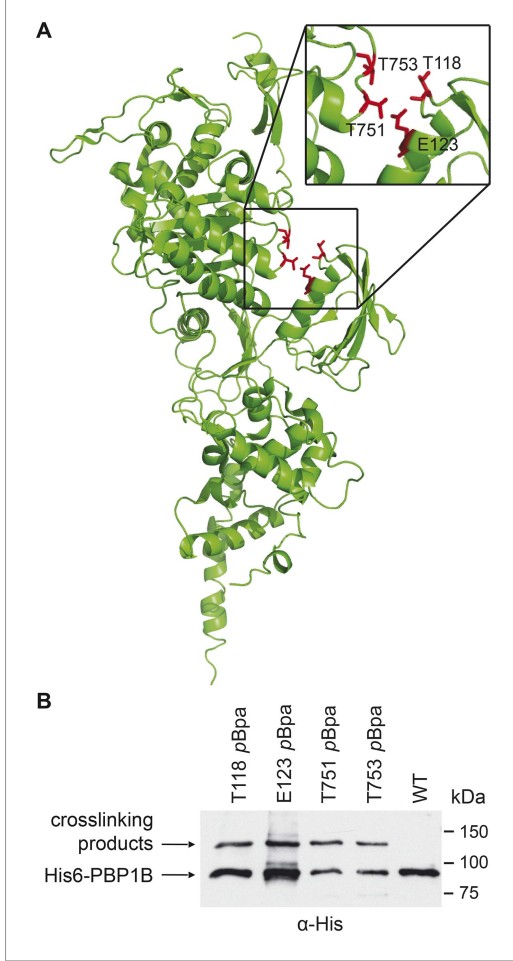

**Figure 3**. PBP1B interacts with CpoB in vivo. (**A**) Photo-cross-linkable amino acid (*p*Bpa) substitutions at the indicated positions in His6-PBP1B, lining a cleft between the TPase and UB2H domains, formed similar UV cross-linking products in vivo. (**B**) SDS-PAGE and α-His6 immunoblot analysis revealed a cross-linked adduct ~120 kDa in size (MW of PBP1Bγ = 88.9 kDa). Analysis of the products by mass spectrometry revealed the presence of CpoB (MW = 28.2 kDa) cross-linked to PBP1B at each indicated position.

envelope constriction in pre-divisional cells (*Figure 4A*) and was positioned preferentially at mid-cell with increasing prevalence as the cell cycle progressed (*Figure 4B,C*). This pattern of localization was corroborated for the native protein, as visualized by immunolabeling cells with antibody specific to CpoB (*Figure 4—figure supplement 2*). Mid-cell positioning of CpoB-mCherry was coincident with envelope constriction (*Figure 4A–D*), suggesting a role during that stage of cell division.

Tol localizes at constriction sites during later stages of cell division (*Gerding et al., 2007*). To better define the timing of CpoB recruitment, we compared the cell cycle dynamics of septal CpoB-mCherry localization with those of other divisome components (*Figure 4B,D*). As cell length correlates with cell cycle progression, measuring fluorescence enrichment at mid-cell in cells sorted by length provides a quantitative metric for temporal comparison (*den Blaauwen et al., 1999*; *Aarsman et al., 2005*; *van der Ploeg et al., 2013*). FtsZ initiates cell division and divisome assembly, with PBP3 and FtsN localizing at progressively later stages (*van der Ploeg et al., 2013*; *Figure 4B,D*; *Figure 4—figure supplement 3*); FtsN is the last essential component recruited in the divisome (*Aarsman et al., 2005*). CpoB-mCherry, a functional TolA-GFP fusion (*Figure 4—figure supplement 1B*), PBP1B, LpoB, and FtsN localized at approximately the same time (*Figure 4B,D*). Thus, CpoB is recruited to the septum during the later stages of cell division, concurrent with Tol, PBP1B, and the onset of OM constriction.

We next asked which proteins and cell division events are required for recruitment of CpoB to the septum. CpoB failed to localize in cells with temperature-sensitive (ts) FtsZ, FtsA, FtsW, or PBP3 under non-permissive conditions, and in cells depleted for FtsN, indicating that CpoB localization is dependent on divisome assembly (*Figure 4—figure supplement 4A–F*). CpoB also failed to localize following inhibition of PBP3 with aztreonam (*Figure 4—figure supplement 4G*), indicating that its localization also requires ongoing septal PG synthesis. However, CpoB localized normally in the absence of PBP1B (*Figure 4—figure supplement 5A*). TolA dependence was more difficult to examine, since delayed OM constriction in the absence of TolA leads to the non-specific accumulation of periplasmic proteins at cell division sites (*Gerding et al., 2007*). Under growth conditions that minimized this effect, CpoB continued to localize in the absence of TolA (*Figure 4—figure supplement 5B*), suggesting that it can be recruited independently of TolA. Because a strain lacking PBP1B and TolA is only marginally viable (*Typas et al., 2010*), we cannot distinguish whether CpoB localization is independent of both PBP1B and TolA, or whether each partner alone is sufficient to drive CpoB localization.

CpoB localization is thus tied to cell division. Further, CpoB interacts and co-localizes with proteins involved in both OM constriction (TolA; *Krachler et al., 2010*; *Gerding et al., 2007*) and PG septal

**Table 1**. His6-PBP1B *p*Bpa—CpoB cross-linking mass spectroscopy data

| Mutant | Score* | Coverage† | Peptides‡ | PSM§ |
|--------|--------|-----------|-----------|------|
| T118 | 300.52 | 19.77% | 7 | 12 |
| E123 | 79.99 | 11.41% | 3 | 4 |
| T751 | 79.99 | 11.41% | 3 | 4 |
| T753 | 301.64 | 19.77% | 7 | 12 |

*Addition of individual scores of the ion fragmentation spectra of the identified peptides.
†Percentage of protein sequence covered by the identified peptides.
‡Number of unique peptides of the protein identified in the sample.
§Number of individual spectra in which the peptides were identified.

synthesis (PBP1B; *Figure 3*), raising the possibility that CpoB participates in coupling these processes following recruitment to the septum during cell division.

## CpoB, TolA, PBP1B, and LpoB form a complex

Interactions between TolA and CpoB (*Krachler et al., 2010*) and between PBP1B and LpoB (*Paradis-Bleau et al., 2010*; *Typas et al., 2010*; *Egan et al., 2014*) have been previously reported. We searched for associations between these complexes that might facilitate functional coordination by testing for further direct pairwise interactions between these proteins in vitro. As measured by surface plasmon resonance (SPR), CpoB bound to immobilized PBP1B with an apparent dissociation constant ($K_D$) of 102 ± 21 nM, but not to PBP1A or to a surface without protein (*Figure 5A,B*), indicating a direct and specific interaction. CpoB is a trimer under the conditions used (*Krachler et al., 2010*), but it is unclear whether CpoB binds as a monomer and/or trimer to PBP1B. PBP1B and TolA also interact directly, as untagged TolA was retained on Ni-NTA beads in the presence but not absence of His6-PBP1B (*Figure 5C*). A purified soluble version of TolA lacking its transmembrane anchor (domain I) was not pulled down with His6-PBP1B (*Figure 5C*), suggesting that domain I of TolA is important for the interaction. In contrast, LpoB does not bind either CpoB or TolA directly, as a His-tagged version of LpoB did not retain CpoB or TolA on Ni-NTA beads, while retaining PBP1B as expected (*Figure 5—figure supplement 1A*).

To contextualize these pairwise interactions, we assessed higher order interactions. As CpoB and TolA each interact with PBP1B, we tested whether they also form a ternary complex. Indeed, chemically cross-linking a mixture of PBP1B, TolA, and CpoB identified a high-molecular weight species containing all three proteins (*Figure 5—figure supplement 1B*). Using this technique, we did not detect analogous high molecular weight complexes of PBP1A with TolA or CpoB (*Figure 5—figure supplement 1B*), confirming that the CpoB and TolA interactions and complex formation with PBP1B were specific. Since LpoB is required for PBP1B activation in vivo, we next tested whether CpoB and TolA interact with PBP1B in complex with LpoB. Indeed, in the presence but not absence of PBP1B, His-LpoB retained CpoB and TolA on Ni-NTA beads after chemical cross-linking, indicating that LpoB-PBP1B-CpoB and LpoB-PBP1B-TolA form ternary complexes (*Figure 5—figure supplement 1A*). Given that all possible ternary complexes containing PBP1B can be formed, we propose that PBP1B, LpoB, CpoB, and TolA also form a quaternary complex.

As CpoB can form a complex with both PBP1B-LpoB and TolA, its cellular abundance might influence which interactions occur in vivo. Using purified protein as a standard and an antibody specific to CpoB, we determined a CpoB cellular copy number of 4550 ± 540 (*n* = 5) in growing cells (*Figure 5—figure supplement 2*). This is similar to an estimated protein synthesis rate for CpoB of ~5200 molecules per generation, determined via ribosome profiling (*Li et al., 2014*). Thus, the abundance of CpoB significantly exceeds the ~520 molecules of PBP1B and ~480 molecules of TolA synthesized per generation (*Li et al., 2014*). CpoB forms trimers, which disassociate to the monomeric form when bound to TolA (*Krachler et al., 2010*). It is unclear in what form(s) CpoB binds to PBP1B. Nevertheless, even in its trimeric form (~1500 trimers), CpoB is in excess to both PBP1B and TolA and is therefore likely able to interact constitutively with both partners.

Taken together, these data suggest that CpoB, PBP1B-LpoB, and Tol associate to form a higher-order complex, spatially linking PG synthesis and OM constriction during cell division.

## CpoB and TolA modulate PBP1B-mediated PG synthesis

We tested the effects of TolA and CpoB on PBP1B GTase and TPase activities in vitro. As measured by consumption of fluorescently labeled lipid II substrate (*Figure 6A* and *Figure 6—figure supplement 1A*),

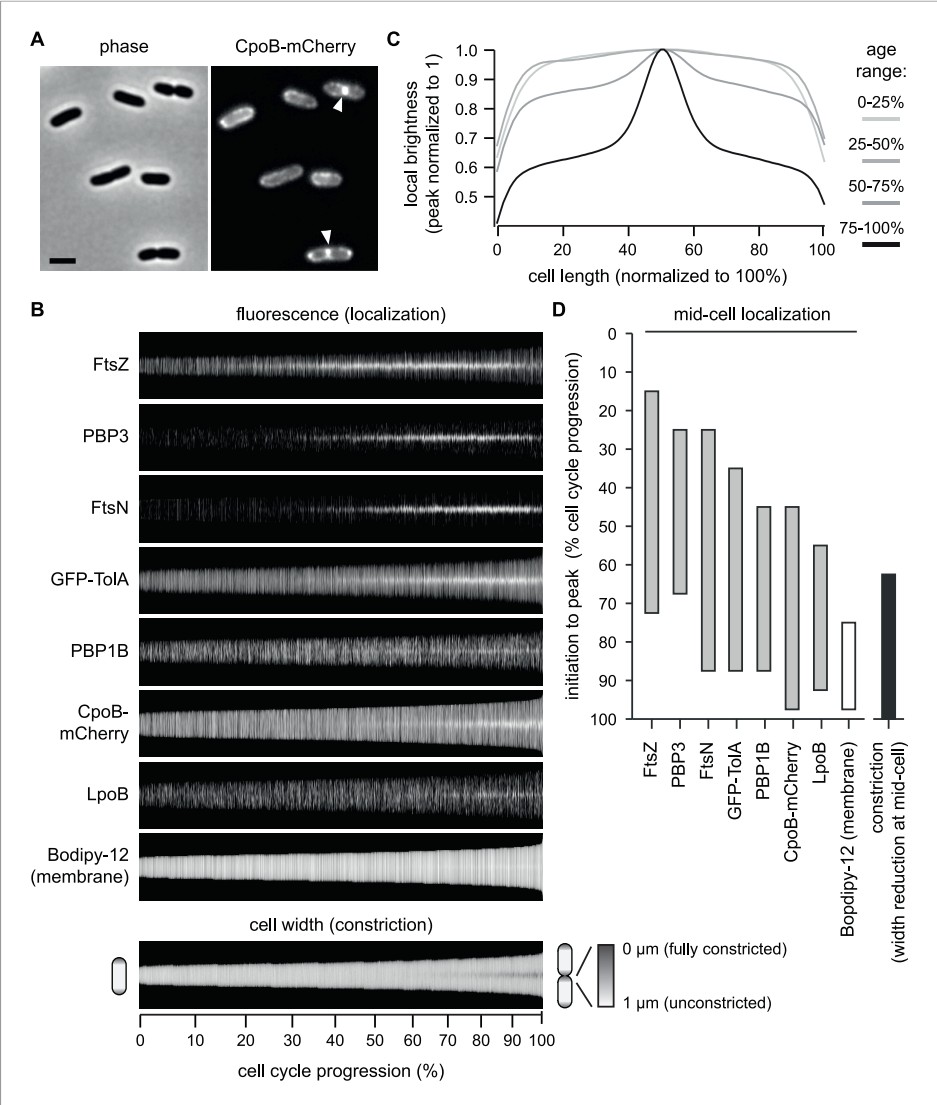

**Figure 4**. CpoB localizes to mid-cell concurrent with PBP1B, TolA, and the onset of OM constriction. (**A**) Endogenously expressed CpoB-mCherry localizes to mid-cell during cell division. Left, phase contrast image; right, mCherry fluorescence; scale bar, 2 μm. (**B**) Localization of CpoB-mCherry as a function of cell cycle progression (cell length), with comparison to other proteins. PBP1B, LpoB, FtsZ, PBP3, and FtsN were visualized by immunofluorescence using specific antibodies. Cell membranes were stained with Bodipy-12. Cell width (diameter) was measured from phase contrast images; constant brightness along the length of the cell indicates constant diameter, and darkening at mid-cell in longer cells indicates constriction. To generate profile maps for each protein, fluorescence intensity profiles (integrated fluorescence as a function of cell length) were derived for >1000 individual cell images and sorted vertically by cell length. Each horizontal line corresponds to a single cell. Y-axis scale indicates % cell-cycle progression based on cell length. Data for FtsZ, PBP3, FtsN, and cell width have been previously published (*van der Ploeg et al., 2013*) and are used here for comparisons. (**C**) Average distribution of CpoB-mCherry for different cell-cycle progression age groups. Fluorescence intensity profiles were normalized by length, then averaged and normalized to the peak local brightness for each age group. (**D**) Relative timing of mid-cell recruitment for each examined protein. Initiation is based on the onset of enriched localization at mid-cell (quantified in *Figure 4—figure supplement 3A,B* and *van der Ploeg et al., 2013*). Peak is the point in the division cycle when maximal localization is reached.

The following figure supplements are available for figure 4:

**Figure supplement 1**. Endogenously-encoded CpoB-mCherry and GFP-TolA fusion proteins are functional.

*Figure 4. continued on next page*

*Figure 4. Continued*

**Figure supplement 2**. CpoB localizes to mid-cell.

**Figure supplement 3**. Timing of localization to mid-cell for CpoB-mCherry, GFP-TolA, and PBP1B, with comparison to divisome proteins FtsZ, PBP3, and FtsN.

**Figure supplement 4**. Localization of CpoB to mid-cell is dependent on divisome assembly and function.

**Figure supplement 5**. Localization of CpoB to mid-cell is independent of PBP1B and TolA.

TolA increased the GTase rate of PBP1B 1.9 ± 0.5-fold, indicating that TolA is a novel regulator of PBP1B. Although weaker than the eightfold stimulation by LpoB (*Figure 6A*; *Egan et al., 2014*), TolA and LpoB stimulations were additive, together yielding an 11.3 ± 0.5-fold increase in GTase activity. These results indicate that LpoB and TolA stimulate PBP1B GTase activity by compatible mechanisms. In contrast, CpoB had no effect on the GTase activity of PBP1B, alone or in combination with LpoB and/or TolA (*Figure 6A*). On the other hand, neither CpoB nor TolA affected basal PBP1B TPase activity (*Figure 6B*, orange), as quantified by percentage of peptides with cross-links in PG produced with radiolabelled lipid II as substrate. Note that, unlike for GTase activity, a continuous TPase assay is currently not available, and therefore we cannot rule out that one of the two proteins affects the TPase rate of PBP1B; this would in fact be expected for TolA as a consequence of GTase activation (see below).

The GTase and TPase activities of PBP1B are coupled, as is generally true for bifunctional synthases (*Bertsche et al., 2005*; *Born et al., 2006*; *Lupoli et al., 2014*). LpoB activates both the PBP1B GTase domain, increasing reaction rate, and the TPase domain, thereby producing hyper-cross-linked PG (*Figure 6B*, blue; *Typas et al., 2010*; *Egan et al., 2014*). Interestingly, although CpoB did not affect the cross-linking activity of PBP1B on its own, it partially prevented the formation of hyper-cross-linked PG in the presence of LpoB (*Figure 6B*, blue). As CpoB affected neither LpoB binding to PBP1B (*Figure 5—figure supplement 1A*) nor GTase stimulation of PBP1B (*Figure 6A*), CpoB must directly interfere with LpoB activation of the PBP1B TPase. Strikingly, TolA alleviated the inhibitory effect of CpoB, restoring the synthesis of highly cross-linked PG in the presence of LpoB (*Figure 6B*, cyan). The effect of CpoB on TPase activity was further enhanced at a moderately increased salt concentration (215 mM NaCl), suggesting that its regulatory effect on PBP1B may be intensified under stress. In this condition, LpoB increased cross-linking from 50% to 60%. The addition of CpoB completely prevented this stimulation, whereas TolA again alleviated the CpoB effect (*Figure 6B*, right graph). Though the magnitudes of these effects are moderate, they are likely significant in vivo given the essentiality of LpoB for PBP1B function under all conditions tested (*Paradis-Bleau et al., 2010*; *Typas et al., 2010*), including different salt concentrations. In summary, interaction of the PBP1B-LpoB synthase complex with TolA further stimulates PBP1B GTase activity, and interactions with CpoB and TolA modulate PBP1B TPase activity via reciprocal effects.

## Regulation of PBP1B by CpoB and TolA responds to Tol energy state in vivo

To establish in vivo relevance and to understand how complex formation and PBP1B activity are regulated in vivo, we systematically characterized interactions between CpoB, TolA, PBP1B, and LpoB via DTSSP cross-linking and co-immunoprecipitation (*Figure 7A–C*). In addition to the previously reported PBP1B-LpoB pair (*Typas et al., 2010*), we observed specific pairwise interactions between CpoB, TolA, and PBP1B, validating that these interactions also occur in vivo. Interestingly, CpoB exhibited minimal or no cross-linking with LpoB in the presence of TolA, but a dramatic increase in cross-linking in its absence (*Figure 7A*). Thus, TolA both prevents CpoB from associating with LpoB (*Figure 7A*) and from interfering with PBP1B TPase stimulation by LpoB (*Figure 6B*), suggesting that these effects are linked. Since an interaction between CpoB and LpoB was not detected in vitro (*Figure 5—figure supplement 1A*), their association in vivo likely occurs in the context of the CpoB-PBP1B-LpoB ternary complex (*Figure 8A*). This possibility could not be tested directly because

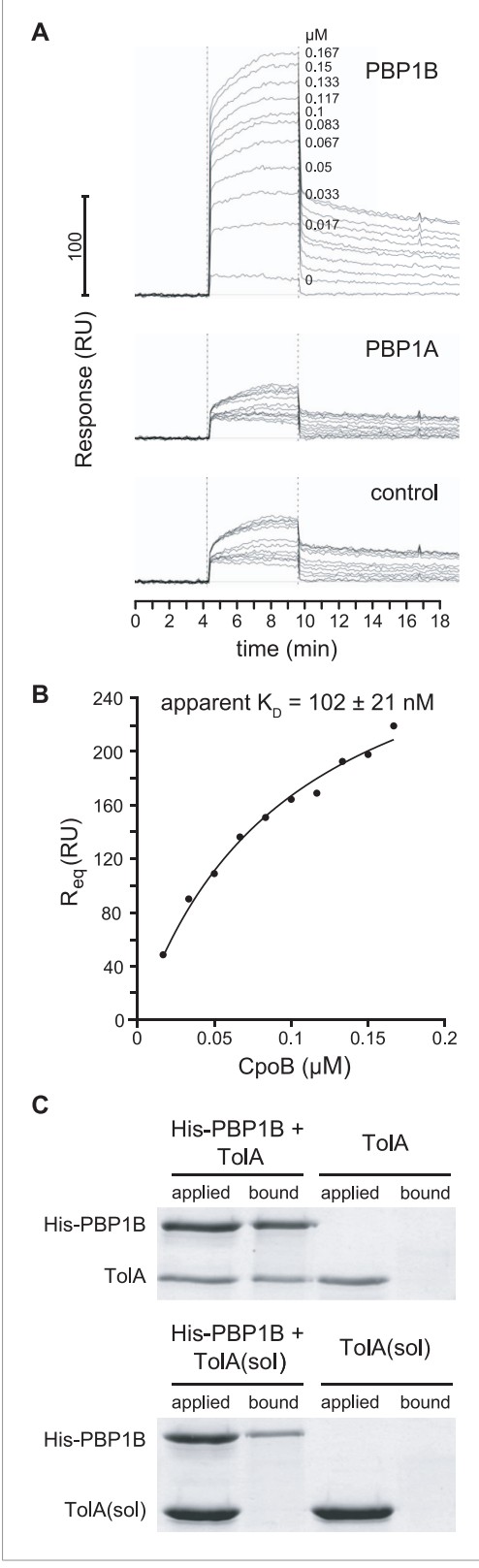

**Figure 5**. PBP1B interacts directly with CpoB and TolA. (**A**) CpoB interacts directly with PBP1B in vitro, assayed by SPR. Sensorgrams show binding of CpoB to a chip surface with immobilized PBP1B, but not PBP1A or an

a strain lacking both PBP1B and TolA is only marginally viable (*Typas et al., 2010*). TolA did not prevent PBP1B from interacting with CpoB (*Figure 7A*); thus, interaction with TolA may instead alter the conformation of CpoB in a manner that both prevents it from interacting with LpoB (*Figure 7A*) and, concurrently, from modulating PBP1B TPase activation by LpoB (*Figure 6B*).

PBP1B-CpoB-TolA complex formation was highly responsive to Tol complex status. In the absence of TolQ, abrogating formation of the TolQR–TolA complex, PBP1B-TolA and PBP1B-CpoB interactions were both substantially diminished. In contrast, CpoB–LpoB interaction and PBP1B-LpoB complex formation increased significantly (*Figure 7A*). Together, these data suggested that CpoB regulates PBP1B-LpoB in response to Tol assembly state, energy state, or both. To distinguish between these possibilities, we compared the effects of a *tolR* deletion, which prevents TolQR–TolA complex assembly, to those of a *tolR*(D23R) point mutation that allows complex assembly but prevents TolQR–TolA from utilizing PMF (*Cascales et al., 2001*). In the *tolR*(D23R) mutant, interaction between TolA and PBP1B was maintained (*Figure 7B*), suggesting that physical association between PBP1B-LpoB and Tol depends on assembly of the TolQR–TolA complex in vivo but not on its energy state. In contrast, interaction between CpoB and LpoB was highly elevated (*Figure 7B*). This strongly suggests that TolQR–TolA must be energized for TolA to prevent CpoB both from interacting with LpoB and, in conjunction, from inhibiting PBP1B TPase stimulation by LpoB. This implies that cycles of TolQR–TolA PMF utilization and energization can in turn modulate PBP1B TPase activity, allowing dynamic regulation of PBP1B activity in response to Tol energy state.

Taken together, interactions between PBP1B-LpoB, TolA, and CpoB promote physical and spatial coordination of PBP1B-LpoB and the Tol apparatus, and provide a mechanism for direct regulation of septal PG synthesis in response to Tol function.

## Discussion

Despite major progress understanding bacterial cell division, it has remained unknown how Gram-negative bacteria coordinate OM invagination with IM invagination and septal synthesis. In this work, we present a mechanism for coordinating PG synthesis and OM constriction, featuring CpoB, previously of unknown function, as its centerpiece. The PBP1B-LpoB and Tol machines

*Figure 5. Continued*

empty control surface. Concentrations of trimeric CpoB (0–0.167 µM) are indicated. (**B**) The $K_D$ of the PBP1B-CpoB interaction was determined by non-linear regression using Sigma Plot 11.0, and assuming that CpoB interacts in its trimeric form. (**C**) TolA interacts directly with PBP1B in vitro only when TolA contains its transmembrane domain I. Interaction between His6-PBP1B and either full length TolA or TolA(sol), a soluble variant lacking domain I, was assessed using an in vitro cross-linking/pull-down approach. His-PBP1B specifically retained TolA, but not the soluble version lacking domain I, when pulled down by Ni-NTA beads. TolA or TolA(sol) alone showed no significant binding to Ni-NTA.

The following figure supplements are available for figure 5:

**Figure supplement 1**. Demonstration of ternary complexes in vitro.

**Figure supplement 2**. CpoB copy number determination.

facilitate PG synthesis and OM constriction, respectively, at the closely-spaced leading and trailing edges of the invaginating envelope (*Figure 1A*). We demonstrate that CpoB connects these processes, promoting physical and functional coordination of the PBP1B-LpoB and Tol machines. We found that both CpoB and TolA interact directly with PBP1B, and our in vitro data support formation of a higher order complex (*Figure 8A*). As depicted in our summary model, interactions between PBP1B, LpoB, CpoB, and TolA allow for dynamic regulation of PBP1B activity, and thus functional coordination of the PBP1B and Tol machines (*Figure 8B*). Below, we first discuss how interaction between these machineries tunes PG synthesis, and then indicate the physiological importance of this process. We end with an evolutionary perspective on these findings.

## How CpoB tunes PBP1B-LpoB function to Tol energy state

In the absence of input from CpoB, LpoB binding to PBP1B stimulates both its GTase and TPase activities. TPase activation results in hyper-cross-linked PG (*Figure 8B*, top left), containing ~70% cross-linked peptides in vitro. CpoB binding interferes only with TPase stimulation (*Figure 8B*, top right) so that ~60% of the peptides are cross-linked. Importantly, this level of cross-linking is similar to that observed in mini-cells, which are formed by abnormal polar divisions in vivo and have an exclusively polar PG (*Obermann and Höltje, 1994*). Therefore, CpoB reduces cross-linkage to a more physiological level. This may be the default cellular state, as CpoB is present in ~10-fold excess over its PBP1B and TolA binding partners (*Figure 5—figure supplement 2*; *Li et al., 2014*). CpoB is the first example of an endogenous regulator that can control the TPase activity of a bifunctional PG synthase independently of its GTase activity. It thus mimics the activity of penicillin G, which blocks PBP1B TPase activity without interfering with GTase stimulation by LpoB (*Lupoli et al., 2014*).

TolA further modulates PBP1B function by reversing CpoB inhibition, in a manner that is dependent on the state of the Tol apparatus (*Figure 8B*, bottom panels). Fuelled by PMF-derived energy, TolQR–TolA cycles between at least two alternative states; we propose that each state has distinct functional consequences for PG synthesis. During the non-energized phase of Tol function, CpoB inhibits LpoB-mediated PBP1B TPase hyper-activation (*Figure 8B*, bottom right). During the energized phase, however, TolA adopts a new conformation that allows it to alleviate CpoB inhibition, thereby restoring maximal TPase stimulation by LpoB (*Figure 8B*, bottom left). Thus, CpoB provides the means to selectively regulate PBP1B catalytic activities, and TolA modulates this effect, linking cycles of PBP1B activity state and the resulting PG cross-linking with cycles of TolQR–TolA energy state while Tol promotes OM invagination.

TolA also directly stimulates PBP1B GTase activity, independently of and additively with LpoB. We do not yet know whether this stimulation occurs during one or both phases of Tol function. Importantly, recruitment and assembly of the TolQR–TolA IM sub-complex is required for regulation of PBP1B by TolA in vivo: in the absence of TolQ or TolR, interaction between TolA and PBP1B is significantly decreased, whereas interaction between PBP1B and LpoB is enhanced. TolQ and TolA localize to the septum independently of each other (*Gerding et al., 2007*), and their recruitment may incorporate different regulatory cues.

## Why is PBP1B tuning necessary for coordinating septal PG synthesis and OM constriction?

We propose that selectively tuning the GTase and TPase activities of PBP1B offers functional flexibility that is particularly important during constrictive PG synthesis. PBP1B-LpoB has the capacity to

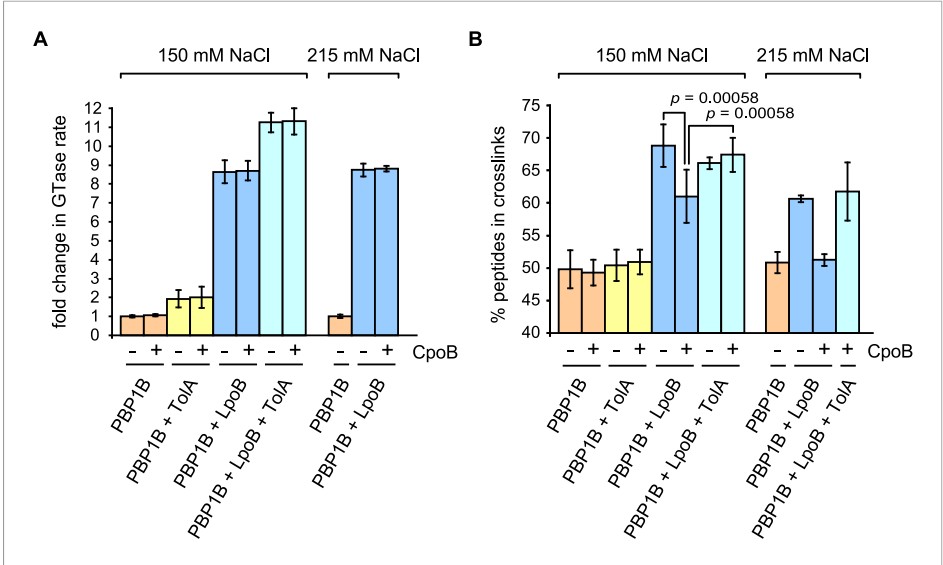

**Figure 6**. CpoB modulates stimulation of PBP1B TPase activity by LpoB. GTase and TPase activities of PBP1B in the presence of LpoB, TolA, and/or CpoB. (**A**) GTase rate was assayed by consumption of fluorescently labelled lipid II by PBP1B in vitro. Change in GTase rate is relative to PBP1B alone and is shown as mean ± SD ($n = 3–4$). (**B**) TPase activity was determined by analyzing peptide cross-linkage following in vitro PG synthesis by PBP1B with radiolabelled lipid II substrate. TPase activity is shown as percentage of peptides present in cross-links in PG produced by PBP1B (mean ± SD). TPase assays were performed at 150 mM NaCl ($n = 4–7$) and at an increased salt concentration of 215 mM NaCl ($n = 2$, shown as the mean ± maximum and minimum values), at which the negative impact of CpoB on PBP1B TPase stimulation by LpoB is exacerbated. Statistical significance/difference was determined by Mann–Whitney U test, with $p$ values of 0.00058 for both data sets.
The following figure supplement is available for figure 6:

**Figure supplement 1**. PBP1B GTase and TPase activity assays.

produce highly cross-linked PG, which may be required at certain stages of new pole synthesis, but the TPase activity of PBP1B is normally down-regulated by CpoB. Up-regulation is allowed only during energized phases of Tol function. Thus, for PBP1B-LpoB at the septal leading edge, increased cross-linking (TPase) activity is permitted only when the OM is brought into close proximity—as facilitated by cycles of Tol function and energy utilization—allowing energized TolQR–TolA to counteract CpoB. The IM-bound septal leading edge can thereby sense and respond the status of the trailing edge of the constricting envelope (i.e. the OM), and activity cycles of the two machines are synchronized, helping to ensure a constant distance between the OM and IM. Restricting TPase activity based on Tol status may also promote better coordination of septal PG synthesis and septal cleavage by OM-controlled amidases (*Uehara et al., 2010*; *Yang et al., 2011*).

By making PBP1B activity responsive to the presence, assembly and energy state of Tol, CpoB enables potential feedback regulation of PG synthesis based on the status of OM invagination. The importance of such coordination can be seen when it is broken, as demonstrated by the phenotypes of a strain lacking CpoB. Multiple antibiotic sensitivities, envelope defects, and genetic interactions paralleling those of a strain lacking PBP1B suggested that CpoB contributes to proper PBP1B function in vivo, an interpretation bolstered by the facts that PBP1B overexpression alleviated these Δ*cpoB* phenotypes and a strain lacking both proteins phenocopied a PBP1B mutant. Envelope defects in the absence of CpoB were relatively mild, likely because redundant mechanisms compensate for loss of coordination by CpoB (e.g., LpoA TPR domain function, discussed below). Yet defects were greatly exacerbated under osmotic stress (*Figure 2—figure supplement 1A*) or in the absence of LpoA function (*Figure 2B–F* and *Figure 2—figure supplement 4B–D*), indicating that coordination by CpoB becomes more crucial under stress and/or when cells must rely more heavily on PBP1B function. Apparent loss of PBP1B function in the absence of CpoB may result from loss of proper coordination

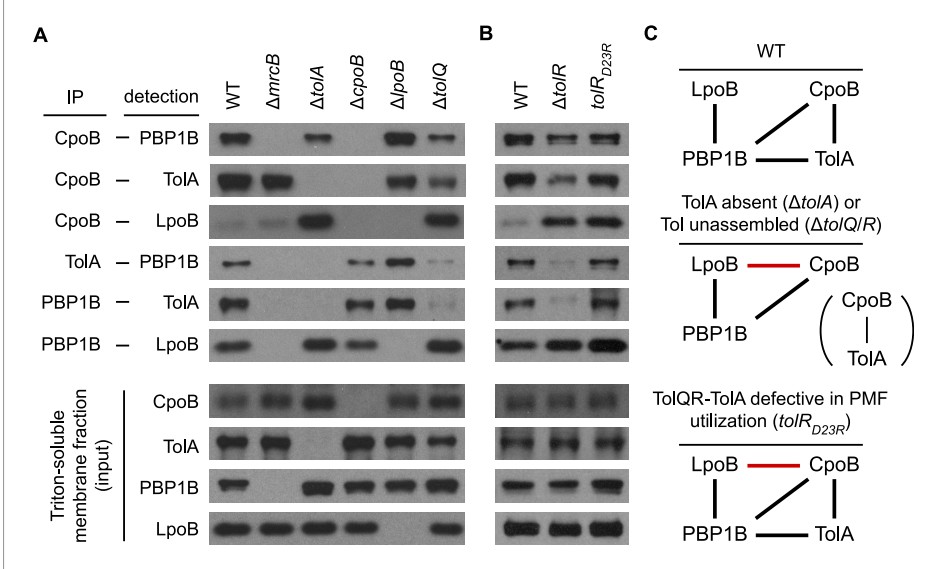

**Figure 7**. Regulatory interactions between CpoB, TolA, and PBP1B-LpoB respond to Tol apparatus assembly and energy state. (**A**) and (**B**). Interactions between CpoB, TolA, PBP1B, and LpoB were characterized in vivo by cross-linking and co-immunoprecipitation for wild type (WT) and indicated mutant strains. Antibodies specific for CpoB, TolA, and PBP1B were used to immunoprecipitate their antigens from Triton X-100-solubilized *E. coli* membrane extracts derived from cells treated with DTSSP cross-linker. Interacting proteins were detected in the immuno-precipitates by western blot using specific antibodies. IP and input panels are from different blot exposures (see 'Methods'). (**C**) Schematic of observed in vivo interactions. Black bars indicate interactions observed in WT; red bars indicate novel interactions in mutant strains.

with Tol that is required for efficient and productive utilization of PBP1B activity during septation. In addition, loss of TPase modulation may itself be deleterious, as uncontrolled PG synthase activity can be as detrimental to the cell as no activity (*Cho et al., 2014*). Synthetic and hydrolytic enzymatic reactions have to be carefully balanced for seamless PG growth, a principle exploited by β-lactam antibiotics, which lyse cells predominantly via uncontrolled PG hydrolase activity (*Kohlrausch and Höltje, 1991*). In this scenario, overexpression of PBP1B may alleviate ΔcpoB phenotypes by ensuring that there is enough unbound (hypoactive) PBP1B in the cell to counter-balance hyperactive PBP1B-LpoB in septal PG synthesis.

Coordination of the PBP1B and Tol machines by CpoB may also be important for proper Tol function. Defects in OM integrity in the absence of PBP1B or CpoB are consistent with such a role. The molecular mechanism by which Tol facilitates OM constriction remains poorly defined (*Gerding et al., 2007*; *Egan and Vollmer, 2013*), and the energized system is currently not amenable to in vitro dissection. It is thus more difficult to determine specific regulatory effects of CpoB and PBP1B on Tol activity. However, coordination likely promotes synchronous OM constriction during cell division, and reciprocal regulation of Tol activity would allow bidirectional coordination of function. Since PBP1B-LpoB can partially compensate for loss of Tol (*Typas et al., 2010*), it is possible that the two complexes contribute to OM constriction cooperatively. Finally, it is possible that both Tol and CpoB play additional roles in PG synthesis and/or other envelope processes, which may be elucidated as we continue to improve our understanding of Tol function in vivo.

## Specificity and modularity in PG synthase evolution

PBP1B and PBP1A retain some core functional redundancy, which allows cell survival when only one is present (*Yousif et al., 1985*). This partial redundancy adds robustness to cell wall biosynthesis. However, PBP1B and PBP1A also have distinct roles. PBP1B is adapted to play a key role in cell division in *E. coli*, as illustrated by phenotypic, localization, and physical interaction data (*Bertsche et al., 2006*; *Muller et al., 2007*; *Typas et al., 2010*), and by the fact that cells without PBP1B lyse

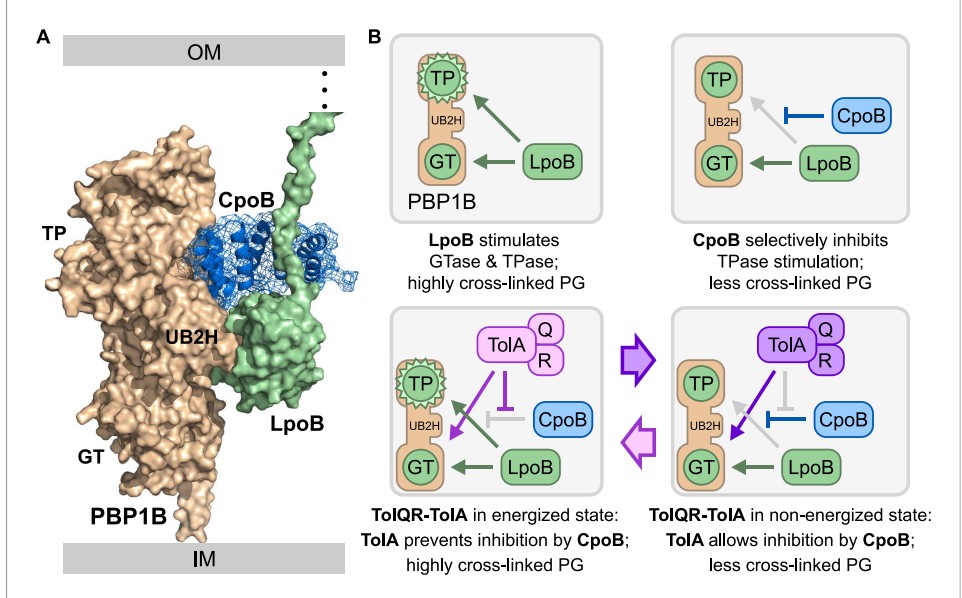

**Figure 8**. Model for physical and functional coordination of the PBPB-LpoB PG synthesis and Tol OM constriction machines by CpoB. (**A**) Data-driven docking of the PBP1B/LpoB/CpoB complex calculated with HADDOCK/CNS protocols (*de Vries et al., 2010*) and integrating experimental data. The lowest energy structure obtained is shown with PBP1B, LpoB, and CpoB colored in wheat, pale green, and blue, respectively. LpoB binds the PBP1B UB2H domain, while CpoB binds in an adjacent cleft between the UB2H and TPase domains (*Figure 3*). We propose that CpoB can contact LpoB in this conformation, preventing hyper-activation of the PBP1B TPase by LpoB. TolA counteracts this CpoB effect, and increases both PBP1B TPase activity (by restoring stimulation by LpoB) and GTase activity (directly). (**B**) Schematic of PBP1B regulation by LpoB, CpoB, and TolA. TP: transpeptidase domain; GT: glycosyltransferase domain. Green circles indicate activity stimulation (increase in glycan synthesis rate for GTase; formation of more highly cross-linked product for TPase). Top left: LpoB binding stimulates both PBP1B GTase and TPase activities; the latter produces highly cross-linked PG. Top right: CpoB modulates TPase stimulation without interfering with GTase stimulation. Bottom left and right: When energized, TolA reverses the effect of CpoB, restoring production of highly cross-linked PG. Regulation by TolA is thus dependent on TolQR–TolA PMF utilization, tying cycles of PBP1B TPase activation to cycles of Tol function.

from the mid-cell at a higher frequency (*Garcia del Portillo and de Pedro, 1990*). Here, we provide additional evidence for the specialized role of PBP1B-LpoB complex in cell division, as it interacts and physically coordinates its function with the Tol apparatus (*Gerding et al., 2007*). Physical and regulatory interactions thus link PBP1B to septal PG synthesis (PBP3, FtsN, FtsW; *Bertsche et al., 2006*; *Muller et al., 2007*; *Fraipont et al., 2011*), PG hydrolysis (MltA; *Vollmer et al., 1999*), and now OM constriction (Tol and CpoB), integrating division processes that span the envelope. Multiple divisome-associated regulators (FtsN, LpoB, CpoB, TolA) either stimulate or inhibit PBP1B activity. Regulation by other factors may be synergistic or antagonistic with regulation by CpoB and TolA, or alternatively may be employed at different stages of septation, helping PBP1B to further address unique requirements of constrictive PG synthesis.

PBPs and other PG-related enzymes (e.g., amidases; *Uehara et al., 2009*) are often characterized by modular architecture, combining broadly conserved catalytic domains with evolutionarily confined non-catalytic domains. We have proposed that these non-catalytic domains provide specificity and diversification to the enzymes (*Typas et al., 2012*). In the case of PBP1B, the non-catalytic UB2H domain shares an extensive binding interface with LpoB (*Egan et al., 2014*), and as we show here is also involved in binding CpoB. Interestingly, LpoB and CpoB approach UB2H from different sides (*Figure 8A*), consistent with their independent binding to PBP1B. CpoB and LpoB come close in this conformation, and this proximity could facilitate the LpoB–CpoB interaction, which is stabilized in the non-energized state of the Tol system. As the UB2H domain and LpoB are present only in enterobacteria and Vibrionaceae (*Typas et al., 2010*; *Dörr et al., 2014*), it remains to be determined

whether and how the more conserved Tol system is integrated with septal PG synthesis in other bacteria. Interestingly, CpoB is not conserved in all bacteria with a Tol system, or even in some that contain LpoB (e.g., Pasteurellaceae).

We postulate that other niche-specific factors will tightly coordinate PG septal synthesis and constriction of the different envelope layers in microbes that lack CpoB and/or LpoB. In this regard, our finding that LpoA has an additional function beyond activating PBP1A, one that is redundant with (or otherwise exacerbates the need for) CpoB and is independent of PBP1A, is particularly interesting. First, it extends for the first time the principle of modularity to PG enzyme regulators, as LpoA has two physically separated domains: one for binding and activating PBP1A (C-terminal domain) and one for its CpoB-related function (N-terminal TPR domain). Second, as both CpoB and LpoA possess TPR domains, which are generally involved in protein–protein interactions, it raises the question of whether both their TPR domains are involved in additional binding and/or regulation with other PG-related enzymes. Such function(s) could be independent of PBP1A/B binding and regulation. An alternative scenario, which would explain the CpoB and LpoA redundancy, is that LpoA can directly link to the Tol system. In contrast to CpoB, we could not detect a direct interaction between LpoA and soluble TolA in vitro. However, the LpoA connection to TolA may be more complex (e.g., dependent on conformational state of TolA), or LpoA may achieve a CpoB-like function via connections to other Tol components. Such a connection would be consistent with the ability of PBP1A to partially substitute for PBP1B, and with the general role of the Tol system in envelope integrity. As all these components (PBP1B/A, LpoB/A, CpoB, and Tol) localize independently to division sites, multiple redundant or alternative paths may exist for coordinating PG synthesis and OM constriction, increasing the robustness of the process to function under different conditions and across different organisms.

## Conclusions

Physical and functional coordination of PBP1B with Tol, facilitated by CpoB, reveals an important additional layer in the regulatory circuitry that controls PBP1B function, and paints an increasingly intricate picture of PBP1B as a regulatory scaffold that is uniquely adapted for the coordination of cell division process in *E. coli*. This is the first mechanistic view of how different envelope layers talk to each other as they grow and constrict during cell division in Gram-negative bacteria. Importantly, the process retains a high degree of modularity, which would allow for individual components to be replaced across evolution, and presumably to some degree in *E. coli* itself. Continued dissection of the dynamic, multi-enzyme, membrane-spanning machines that mediate coordinated cell division processes—including investigation of which interactions occur simultaneously, when and where in the cell they occur, and what additional regulatory relationships they confer—will provide greater understanding of the molecular mechanisms that govern interconnected divisome functions and their synchronization. Looking across different organisms will also provide a better overview of which interconnections are broadly required and by what different means they can be achieved. Considering the many facets of cell envelope composition (*Radolf et al., 2012*) and cell division (*Leisch et al., 2012*), these questions are bound to have different answers in different bacteria. Nevertheless, the geometric and topological changes inherent during cell division likely generally necessitate a high degree of feedback between the different envelope layers through physical connections that modulate biochemical activities (*Weiss, 2015*).

## Materials and Methods

### Chemicals and proteins

[$^{14}$C]GlcNAc-labelled lipid II and dansylated lipid II were prepared as published (*Breukink et al., 2003*; *Bertsche et al., 2005*). The following proteins were prepared as previously described: PBP1B (*Bertsche et al., 2006*), His-LpoB(sol), and LpoB(sol) (*Egan et al., 2014*). Antisera against PBP1B, LpoB, CpoB, TolA, and TolB (rabbit) were obtained from Eurogentec (Liege, Belgium) and purified over an antigen column as described (*Bertsche et al., 2006*). Cellosyl was provided by Hoechst AG (Frankfurt, Germany). VIM-4 was a kind gift from Adeline Derouaux.

### Protein overproduction and purification

BL21(DE3) strains harbouring plasmids pET28-His6-CpoB (for the purification of CpoB; signal sequence replaced by a cleavable oligohistidine tag), pET28-TolA-His6 (for the purification of full length TolA with a C-terminal oligohistidine tag), pET28-His6-TolA (for the purification of full length

TolA with a cleavable N-terminal oligohistidine tag) or pET28-His6-TolA(sol) (for purification of TolA (72-421)) were grown in 3 L of LB medium with appropriate supplements at 30°C to an $OD_{578}$ of 0.5–0.6. Recombinant genes were overexpressed by adding 1 mM IPTG to the cell culture followed by a further incubation for 3 hr at 30°C. Cells were harvested by centrifugation (10,000×$g$, 15 min, 4°C) and the pellet was resuspended in buffer I (25 mM Tris/HCl, 10 mM $MgCl_2$, 500 mM NaCl, 20 mM imidazole, 10% glycerol, pH 7.5). A small amount of DNase, protease inhibitor cocktail (Sigma-Aldrich, St. Louis, MO; 1/1000 dilution), and 100 µM phenylmethylsulfonylfluoride (PMSF) were added before cells were disrupted by sonication (Branson digital). The lysate was centrifuged (130,000×$g$, 1 hr, 4°C). At this point, purification procedures of full length TolA constructs and of CpoB and TolA(sol) differ.

For CpoB and TolA(sol), the supernatant was applied to a 5-mL HisTrap HP column (GE healthcare Bio-Sciences, Piscataway, NJ), attached to an ÄKTA Prime+ (GE Healthcare Bio-Sciences) at 1 mL/min. The column was washed with 4 volumes buffer I before step-wise elution of bound proteins with buffer II (25 mM Tris/HCl, 10 mM $MgCl_2$, 500 mM NaCl, 400 mM imidazole, 10% glycerol, pH 7.5). To remove the oligohistidine tag from His-CpoB and His-TolA(sol), 50 U/mL of restriction grade thrombin (Merck Millipore, Darmstadt, Germany) was added, and the protein was then dialyzed against 2 L of 25 mM Tris/HCl, 10 mM $MgCl_2$, 500 mM NaCl, 10% glycerol, pH 7.5 for 18 hr at 4°C. Proteins samples were then concentrated to 4–5 mL using a VivaSpin-6 column (MW cut-off 6000 Da) and applied to a Superdex200 HiLoad 16/600 column at 0.8 mL/min for size exclusion chromatography in 25 mM Tris/HCl, 1 M NaCl, 10% glycerol, pH 7.5. Finally, proteins were dialyzed against storage buffer (25 mM Tris/HCl, 500 mM NaCl, 10% glycerol, pH 7.5).

For full length TolA or TolA-His, the membrane pellet resulting from the above ultracentrifugation was resuspended in extraction buffer (25 mM Tris/HCl, 10 mM $MgCl_2$, 1 M NaCl, 2% Triton X-100, 10% glycerol, pH 7.5) and incubated overnight with mixing at 4°C. Samples were centrifuged (130,000×$g$, 1 hr, 4°C) and the supernatant applied to a 5-mL HisTrap HP column (GE Healthcare Bio-Sciences) attached to an ÄKTA Prime+ (GE Healthcare Bio-Sciences) at 1 mL/min. The column was washed with 4 volumes extraction buffer, followed by 4 volumes of wash buffer I (25 mM Tris/HCl, 10 mM $MgCl_2$, 1 M NaCl, 20 mM imidazole, 2% Triton X-100, 10% glycerol, pH 7.5), followed by a final wash with 4 volumes of wash buffer II (as wash buffer I, with 40 mM imidazole and 0.2% Triton X-100). Bound protein was eluted step-wise with elution buffer (25 mM Tris/HCl, 10 mM $MgCl_2$, 500 mM NaCl, 400 mM imidazole, 0.2% Triton X-100, 10% glycerol, pH 7.5). At this point, TolA-His was dialyzed into storage buffer (25 mM Tris/HCl, 500 mM NaCl, 0.2% Triton X-100, 10% glycerol, pH 7.5). To remove the oligohistidine tag from His-TolA, 50 U/mL of restriction grade thrombin (Merck Millipore) was added to the protein, which was then dialyzed against 2 L of 25 mM HEPES/NaOH, 10 mM $MgCl_2$, 50 mM NaCl, 10% glycerol, pH 6.0 for 24 hr at 4°C. Sample was then applied to a 5-mL HiTrap HP SP column attached to an ÄKTA Prime+ (GE Healthcare Bio-Sciences) for ion exchange chromatography, at a flow rate of 0.5 mL/min in buffer A (25 mM HEPES/NaOH, 10 mM $MgCl_2$, 50 mM NaCl, 0.2% Triton X-100, 10% glycerol, pH 6.0). The column was washed with 6 volumes buffer A before stepwise elution of bound TolA with buffer B (as A, with 500 mM NaCl). TolA sample was then dialyzed into storage buffer (25 mM HEPES/NaOH, 10 mM $MgCl_2$, 100 mM NaCl, 0.2% Triton X-100, 10% glycerol, pH 7.5). Recombinant PBP1B (pDML924; the functional, short PBP1Bγ version, starting from amino acid 46) was purified as described previously (*Bertsche et al., 2005*). Versions of proteins retaining their His-tags were purified as above, omitting the addition of thrombin.

## Bacterial strains, plasmids, and growth conditions

Bacterial strains and plasmids used in this work are listed in *Supplementary File 1A*. Primers used in this work are listed in *Supplementary File 1B*.

## Growth conditions

For in vivo assays, cells were grown aerobically at 30°C or 37°C in Lennox Luria–Bertani (LB) medium (10 g/L tryptone, 5 g/L yeast extract, 5 g/L NaCl) (Fisher Scientific, Houston, TX) unless otherwise indicated. Where appropriate, antibiotics or inducers were added: ampicillin (100 µg/mL), chloramphenicol (10–34 µg/mL), kanamycin (30 µg/mL), arabinose (0.2–1%, wt/vol), IPTG (1 mM). For protein production, cells were grown aerobically at 30°C or 37°C in Miller LB medium (10 g/L tryptone, 5 g/L yeast extract, 10 g/L NaCl).

## Construction of *E. coli* strains

To generate and combine *E. coli* gene deletions, *kan*-marked alleles from the Keio *E. coli* single-gene knockout library (*Baba et al., 2006*) were transferred into relevant background strains using P1 phage transduction (*Thomason et al., 2007*). The Keio pKD13-derived *kan* cassette is flanked by FRT sites, allowing removal of the *kan* marker via expression of FLP recombinase to generate unmarked (kanamycin-sensitive) deletions with a FRT-site scar sequence (*Datsenko and Wanner, 2000*; *Baba et al., 2006*).

Other chromosomal mutations and gene fusions (*lpoA*(Δ58-252), *tolQ*(D23R), *cpoB-mCherry*, and *gfpmut2-tolA*) were generated via lambda Red recombinase-mediated oligonucleotide and/or PCR recombineering (*Thomason et al., 2014*) using recombineering plasmid pSIM19 (*Datta et al., 2006*). Desired mutations were confirmed by PCR and sequencing. To generate the *lpoA*(Δ58-252) strain, a *sacB-kan* cassette from pIB279 (*Blomfield et al., 1991*) was amplified using primers ANG014 x ANG015 and inserted into *lpoA* to generate precursor strain CAG70134 (BW25113 *lpoA*(Δ1-255:: (*sacB-kan*)). A PCR product containing the *lpoA*(Δ58-252) allele was then generated via overlap extension ('stitching') PCR (*Heckman and Pease, 2007*), using PCR primers ANG235 x ANG239 and ANG241 x ANG242 with *E. coli* genomic DNA as template for the first round of PCR and ANG235 x ANG242 with first-round-PCR-products as template for the second round of PCR. The PCR product was then used to replace the *sacB-kan* cassette via counterselection on no-salt LB plates (10 g/L tryptone, 5 g/L yeast extract, 18 g/L agar) containing 7% (wt/vol) sucrose. To generate the *tolQ* (D23R) strain, a *sacB-cat* cassette from plasmid pDS132 (*Philippe et al., 2004*) was amplified using primers ANG478 x ANG479 and inserted into *tolQ* to generate strain CAG70764 (BW25113 *tolR* (D23::(*sacB-cat*)). The *tolQ*(D23R) mutation was then introduced by using oligonucleotide ANG484 to replace the *sacB-cat* cassette via sucrose counterselection. To generate the *cpoB-mCherry* strain, sacB-cat was inserted after *cpoB* using primers ANG143 x ANG144 and replaced with a *GSGSGSGS* linker followed by *mCherry* using primers ANG177 x ANG178 with pMG36 (encodes *pal-mCherry*) (*Gerding et al., 2007*) as template. To generate the *gfpmut2-tolA* strain, *sacB-cat* was inserted before *tolA* using primers ANG263 x ANG264 and replaced with gfpmut2 followed by an *ATGTRT* linker using primers ANG347 x ANG349 with pBAD24-*gfp* (*Nilsen et al., 2004*) (encodes *gfpmut2*) as template.

To transfer chromosomal mutations and gene fusions to other strain backgrounds, the *sacB-cat* or *sacB-kan* cassette was first transferred to the intended recipient strain by P1 phage transduction (*Thomason et al., 2007*). The unmarked mutant allele was then transferred into the *sacB-cat* or *sacB-kan*-bearing recipient strain by P1 phage transduction with selection on no-salt LB plates containing 7% (wt/vol) sucrose and 10 mM potassium citrate. (Potassium citrate is substituted for sodium citrate because the addition of sodium prevents *sacB*-mediated sucrose sensitivity.)

## Construction of plasmids

pBAD33-PBP1B was constructed by cloning *mrcB* (encodes PBP1B) along with 20 bp of upstream sequence so as to include the native ribosome binding site (PCR primers: ANG093 x ANG094) into the SacI/XbaI restriction sites of pBAD33 (*Guzman et al., 1995*). Similarly, pBAD33-CpoB was constructed by cloning *cpoB* with 25 bp upstream sequence into SacI/SphI pBAD33 (PCR primers: ANG005 x ANG006). Similarly, pTolA (pBAD33-TolA) was constructed by cloning *tolA* with 20 bp upstream sequence into SacI/XbaI pBAD33 (PCR primers: ANG147 x ANG148). For pET28-His6-CpoB, *cpoB* was cloned into NdeI/SacI pET28a (EMD Millipore, Billerica, MA) (PCR primers: ANG182 x ANG184). For pET28-His6-TolA, *tolA* was cloned into NdeI/SacI pET28a, replacing the native GTG *tolA* start codon with the NdeI site ATG. For pET28-His6-TolA(sol), *tolA*(74-421) was cloned into NdeI/SacI pET28a. For pET28-TolA-His6, *tolA* (without stop codon) was cloned into NcoI/XhoI. The forward primer used to amplify *tolA* contained a BsaI restriction site, GGTCTCTCATG, to allow cloning into the pET28 NcoI site while preserving the native *tolA* sequence after the start codon.

Amber (TAG) mutations were introduced in pDML924 using the QuickChange PCR site-directed mutagenesis kit and protocol (Stratagene/Agilent Technologies, Santa Clara, CA). PCR reactions were treated with DpnI (Fermentas/Thermo Fisher Scientific, Waltham, MA) to remove parent plasmid and then transformed into *E. coli* DH5α. Candidate plasmid derivatives containing amber mutations were purified from isolated transformants and confirmed by sequencing.

### Growth fitness assays (related to *Figure 2*, *Figure 2—figure supplements 2–4*, and *Figure 4—figure supplement 1*)

*E. coli* gene deletion alleles were derived from the Keio collection (*Baba et al., 2006*). Strains for overexpressing PBP1B carried arabinose-inducible plasmid pBAD33-PBP1B. Strains for expressing YbgF in trans (for complementation of Δ*cpoB*) carried arabinose-inducible plasmid pBAD33-CpoB. *E. coli* wild type and mutant strains were arrayed and grown in a 1536-colony format (n ≥ 96 for each strain) on LB Lennox plates (20 g/L agar) at 37°C, then replica-pinned robotically (Rotor HDA, Singer Instruments, Roadwater, UK) to LB Lennox and/or indicated condition plates. Assay plates were incubated for 8–14 hr at 37°C and then imaged. Colony sizes were determined using in-house software.

### Analysis of periplasmic leakage and OM blebbing (related to *Figure 2* and *Figure 2—figure supplement 4*)

#### Imaging

For DIC and fluorescence imaging for example images (*Figure 2C*), cells were imaged at the Nikon Imaging Center (NIC) at UCSF on a Ti-E Microscope with a Plan Apo VC 100×/1.4 Oil (DIC N2 / 100X I) 65.4 nm/pixel objective. Images were collected using a Nikon Coolsnap HQ2 camera and NIS-Elements v4.2 software (Nikon Instruments Inc., Melville, NY). For phase-contrast and fluorescence imaging for quantitation (*Figure 2D,E* and *Figure 2—figure supplement 4B,C*), cells were imaged on a Nikon Eclipse TE inverted fluorescence microscope with a 100× (NA 1.40) oil-immersion objective. Images were collected using an Andor DC152Q sCMOS camera (Andor Technology, South Windsor, CT) and MicroManager v1.3 software (*Edelstein et al., 2010*).

#### Quantitation of periplasmic leakage and OM blebbing

Strains were grown in LB Lennox broth with 1 mM ITPG to induce periplasmic mCherry expression. 1% arabinose and 10 µg/mL chloramphenicol were added for plasmid-bearing strains. Overnight cultures were diluted 1:200 and then grown in a 96-well microplate (Model #780271; Greiner Bio-One, Monroe, NC) for 2 hr before imaging. Phase-contrast images were segmented using a custom MATLAB software package called Morphometrics, as in *Ursell et al. (2014)*. The software is currently in beta version and freely available upon request (kchuang@stanford.edu); the release version will be made available on Bitbucket. Related scripts for measurement of periplasmic leakage (peripheral fluorescence) and outer membrane blebs/vesicles (fluorescent puncta) are also available from Bitbucket in the kchuanglab/Gray_eLife_CpoB repository. To measure fluorescence profiles along cell outlines, fluorescence intensity was measured along segmented cell contours with a spacing of 1 pixel (0.064 µm). Cells were defined as exhibiting periplasmic leakage if their median contour fluorescence fell below a threshold defined as the 98th percentile of Δ*tolA* cell contour fluorescence. To identify blebs, we first located all fluorescent puncta with average brightness above the 99.7th percentile of WT cell brightness, and with an area smaller than 4 µm². Blebs were then defined as isolated bright puncta within 0.3 µm of a segmented cell.

### Chlorophenyl red-β-D-galactopyranoside (CPRG) penetration assay (related to *Figure 2—figure supplement 1A,B*)

The CPRG assay was performed as described (*Paradis-Bleau et al., 2014*) with slight modifications. Instead of normal rectangular agar plates in which colonies are arrayed next to each other and color can diffuse, we used 384-well plates, filled with LB-agar-CPRG by a liquid handling robot (Biomek FX; Beckman Coulter, Brea, CA). Mutants carrying pCB112, a mobile plasmid encoding *lacZ* under control of the lactose promoter ($P_{lac}$), were arrayed robotically on each well (ROTOR, Singer Instruments). Color development (CPR-red; 570 nm) was monitored for 48 hr (every 30 min) in a multi-well plate reader (Tecan M1000 Pro with a stacker) at room temperature. Absorption wavelength (not to overlap with colony development), agar volume per well and a number of other technical parameters were optimized for the assay (George Kritikos and Athanasios Typas; unpublished data). The accumulation rate of CPR color was calculated after fitting a linear curve on an absorption-time plot.

### Filtered culture supernatants (related to *Figure 2—figure supplement 1C*)

Culture supernatants were collected and filtered using a 0.2 µm-pore SFCA syringe filter (Fisher Scientific), then TCA-precipitated and prepared for analysis by SDS-PAGE and western blot as described (*Wagner et al., 2009*).

## In vivo *p*Bpa cross-linking experiments (related to *Figure 3*)

Amber mutations were introduced at indicated sites via site-directed mutagenesis of plasmid pDML924, which encodes His6-tagged PBP1B (*Terrak et al., 1999*). *E. coli* BL21(DE3) cells were co-transformed with pSup-BpaRS-6TRN (*Ryu and Schultz, 2006*) and either pDML924 or amber mutant derivatives. Cells were grown to an $OD_{600}$ of 0.5–0.6; protein production was then induced with 10 µM IPTG, and 1 mM freshly-prepared *p*-benzoyl-L-phenylalanine (Bachem) dissolved in 1 M NaOH was added. After 2 hr of protein production, cells were harvested by centrifugation, washed with PBS and resuspended in 3 mL PBS, then transferred to a petri dish and exposed to UV light (365 nm) for 1.5 min (photoMax Housing 200W, model 60100, 30 cm distance to sample; Oriel Instruments). Cells were cooled on ice during UV illumination. Cell samples were separated by SDS-PAGE (8%), transferred to a nitrocellulose membrane (Bio-Rad Laboratories, Hercules, CA), and analyzed by western blot using monoclonal anti-polyhistidine peroxidase-conjugated antibody (1:4000 dilution; Sigma-Aldrich).

## Mass spectrometry and related data analysis (related to *Figure 3* and *Table 1*)

Excised protein bands were reduced with DTT, alkylated with iodoacetamide and in-gel digested with trypsin (*Shevchenko et al., 2006*). Nanoflow liquid chromatography coupled to mass spectrometry was performed on an Agilent 1200 nanoflow system (Agilent Technologies) connected to a MS LTQ-Orbitrap XL (Thermo Fisher Scientific). The samples were trapped on a 20 mm ReproSil-Pur C18-AQ (Dr. Maisch GmbH, Ammerbuch, Germany) trapping column (packed in-house, i.d., 100 µm; resin, 5 µm) with a flow-rate of 5 µL/min. Sequential elution of peptides was accomplished using an analytical column (Dr. Maisch GmbH; packed in-house, i.d., 50 µm; resin, 3 µm) with a 35 min gradient of 10–38% buffer B (buffer A, 0.1 M acetic acid; buffer B, 0.1 M acetic acid, 80% [vol/vol] acetonitrile) followed by 38–100% B in 3 min, 100% B for 2 min. The flow rate was passively split from 0.45 mL/min to 100 nL/min (*Nesvizhskii et al., 2003*). Nanospray was achieved using a distally coated fused silica emitter (made in-house, o.d., 375 µm; i.d., 20 µm) biased to 1.7 kV. Mass spectrometer was operated in the data dependent mode to automatically switch between MS and MS/MS. The high resolution survey full scan was acquired in the orbitrap from m/z 350 to m/z 1500 with a resolution of 30.000 (FHMW). The most intense ions at a threshold of above 500 were fragmented in the linear ion trap using collision-induced dissociation at a target value of 10,000.

Peak lists were generated from the raw data files using the Proteome Discoverer software package version 1.3.339 (Thermo Fisher Scientific). Peptide identification was performed by searching the individual peak lists against a concatenated target-decoy database containing the *E. coli* sequences in the Uniprot database (release 2012_06) supplemented with a common contaminants database using the Mascot search engine version 2.3 (Matrix Science, London, United Kingdom) via the Proteome Discoverer interface. The search parameters included the use of trypsin as proteolytic enzyme allowing up to a maximum of two missed cleavages. Carbamidomethylation of cysteines was set as a fixed modification, whereas oxidation of methionines was set as a variable modification. Precursor mass tolerance was initially set at 50 ppm, while fragment mass tolerance was set at 0.6 Da. Subsequently, the peptide identifications were filtered for an ion score of 20.

## Fluorescence localization, immunolocalization, and related image analysis (related to *Figure 4* and *Figure 4—figure supplements 2–5*)

Cells were grown to steady state in glucose minimal medium (GB1) pH 7.0 (*den Blaauwen et al., 1999*) supplemented with 50 µg/mL of required amino acids (LMC500 and derived strains) or 20 µg/mL thymine (BW25113 and derived strains) at 28˚C, or in Lennox LB medium pH 7.0 at 28˚C, as indicated. Absorbance was measured at 450 nm (GB1) or 600 nm (Lennox LB) with a Biochrom Libra S70. Generation times of the LMC500 strain grown in GB1 and Lennox LB medium at 28˚C are 85 and 40 min, respectively, and for BW25113 are 92 and 40 min, respectively.

### Fluorescence localization of CpoB-mCherry and GFPmut2-TolA

*mCherry* was placed in frame after the endogenous *cpoB* gene in the chromosome of *E. coli* MC4100 (LMC500); *gfpmut2* was placed in frame before the *tolA* gene (see 'Bacterial strains and plasmids', above). Cells were grown to steady state in glucose minimal medium at 28˚C, then placed on a 0.5 × 0.5 cm 1% agarose patch (*Koppelman et al., 2004*) containing minimal medium on an object glass to allow for optimal aeration, covered by a cover glass and immediately imaged as described

(*van der Ploeg et al., 2013*). Images were acquired with Micro-Manager (http://www.micro-manager.org/) with direct output of the desired hyperstack structure for ImageJ by Wayne Rasband (http://imagej.nih.gov/ij/). CpoB-mcherry was used because the localization of freely diffusing periplasmic proteins was observed to be perturbed by fixation of cells grown in minimal medium, likely due to osmotic effects. GFP-TolA was used because available antiserum was not useable for immunolabeling. Both fusions were functional (*Figure 4—figure supplement 1*).

## Immunolocalization

*E. coli* cells of wild type strain LMC500 were grown to steady state in glucose minimal medium at 28°C, then fixed while shaking in growth medium as described (*den Blaauwen et al., 2003*). Cells were permeabilized and immunolabelled with indicated antibodies as described (*den Blaauwen et al., 2003*), immobilized on 1% agarose and imaged as above. Bodipy-12, a membrane binding dye, was used as a control to show at what stage constriction results in increased mid-cell fluorescence due to the presence of two membranes. Antisera against PBP1B and LpoB were affinity purified and pre-incubated with cells of their respective deletion strains before use in immunolabeling. Antisera against FtsZ and FtsN were specific without further purification. The immunolabeling method used here has been shown to be reproducible and to not perturb the localization of membrane and cytoplasmic proteins (*van der Ploeg et al., 2013*).

## Related image analysis

Image analysis was performed as described (*van der Ploeg et al., 2013*). Briefly, individual cells were identified and analyzed using the public domain program Coli-Inspector, running under plugin ObjectJ written by Norbert Vischer (University of Amsterdam, https://sils.fnwi.uva.nl/bcb/objectj/), which runs in combination with ImageJ. Via this analysis, parameters including cell length, cell diameter (width) as a function of length, and integrated fluorescence as a function of length (fluorescence profile) were quantified for each cell. For each examined protein, fluorescence profiles of 3000–5000 individual cells were sorted according to cell length to generate profile maps (*Figure 4B*), similar to demographs (*Hocking et al., 2012*). A cell diameter profile map (*Figure 4B*, bottom panel) was generated from imaged cells that were fixed only. Percentage of cell cycle progression (age) for individual cells was determined based on relative cell length ranking, assuming a logarithmic growth of cell length (*Aarsman et al., 2005*). Average fluorescence profiles for either all cells or specific age ranges were generated and plotted against normalized cell length as described (*Potluri et al., 2010*; *Figure 4C*). Cell ages corresponding to the initiation of mid-cell localization and peak mid-cell localization for each examined protein (*Figure 4D*) were determined as described in *Figure 4—figure supplement 3* and below.

## CpoB immunolocalization and related image analysis (related to *Figure 4—figure supplements 2, 4, and 5*)

To examine cell division protein dependencies for CpoB mid-cell localization (*Figure 4—figure supplement 4*), isogenic strains based on the parental strain LMC500 (see 'Bacterial strains and plasmids', above) with temperature-sensitive alleles of cell division proteins (*Taschner et al., 1988*) were used. Cells were grown in glucose minimal medium at 28°C (permissive temperature) to an $OD_{450}$ of 0.2. Subsequently, the growing cells were diluted 1:4 into pre-warmed medium (28°C or 42°C) and grown until the cultures again reached an $OD_{450}$ of 0.2. Cells were fixed, permeabilized, and immunolabeled as described ('Methods' and *den Blaauwen et al., 2003*) using antibody to CpoB that had been pre-adsorbed on permeabilized Δ*cpoB* cells to remove cross-reacting antibodies, as described previously (*Buddelmeijer et al., 2013*). Cells were sorted according to length to generate fluorescence profile maps, as described in 'Methods' (~500–1000 cells per profile). The FtsZ(ts) strain LMC509 (*Taschner et al., 1988*) is not completely temperature sensitive; consequently, CpoB still localizes at mid-cell in shorter cells in this strain; however, in long smooth cells, no mid-cell localization is seen. The PBP3(ts) strain LMC510 (*Taschner et al., 1988*) is already somewhat elongated at the permissive temperature due to the presence of an unstable PBP3 molecule (*Fraipont et al., 2011*). The FtsW(ts) strain JLB17 is not isogenic to LMC500 (*Ishino et al., 1989*; *Mohammadi et al., 2014*) and was grown in Lennox LB medium without salt and with 1% glucose and 20 mg/mL of thymine at 28°C to an $OD_{600}$ of 0.3.

Strain JOE565 (FtsN depletion) (*Chen and Beckwith, 2001*) was grown in Lennox LB with 0.2% arabinose at 28°C to an $OD_{600}$ of 0.3. Cells were washed in warm medium without arabinose and resuspended in medium without arabinose to deplete FtsN. For aztreonam treatment (inhibits PBP3), LMC500 was grown in glucose minimal medium at 28°C to an $OD_{450}$ of 0.2. The culture was then split and diluted 1:4 into medium with or without 2 µg/mL aztreonam and grown for an additional two mass doublings.

To examine dependency on PBP1B or TolA for CpoB mid-cell localization (*Figure 4—figure supplement 5*), wild type strain BW25113 and isogenic derivatives lacking *mrcB* or *tolA* were used. Cells were grown in Lennox LB medium at 28°C to an $OD_{600}$ of 0.3, diluted 1:25 into pre-warmed medium and grown until the cultures again reached an $OD_{600}$ of 0.3, then fixed, immunolabeled with antibody to CpoB and imaged as described above. Cells were sorted according to length to generate fluorescence profile maps, as described in 'Methods' (~1500 cells per profile). The Δ*tolA* strain had increased CpoB immunofluorescence levels, with accumulation at constriction sites (*Figure 4—figure supplement 5B*, middle two panels); TolA overexpression (pTolA; see 'Bacterial strains and plasmids') alleviated this effect and reduced CpoB immunofluorescence levels to below that of wild type (*Figure 4—figure supplement 5B*, right two panels).

## Initiation and peak fluorescence localization (related to *Figure 4—figure supplement 3*)

Age of localization initiation was defined as the age class when extra mid-cell fluorescence was first observed, as shown and described in *Figure 4—figure supplement 3A*. Age of peak localization was defined as the Moment (*Figure 4—figure supplement 3B*), i.e. the age when extra mid-cell fluorescence reached its maximum. For each analyzed cell image, extra mid-cell fluorescence was calculated as 'FCplus' and plotted vs cell age to determine the Moment, as described (*van der Ploeg et al., 2013*).

## In vitro protein interaction and activity assays (related to *Figure 5*, *Figure 5—figure supplement 1*, *Figure 6* and *Figure 6—figure supplement 1*)

SPR experiments were performed as previously described (*Egan et al., 2014*). The concentration of CpoB injected ranged from 0.05 to 0.5 µM, or 0.017–0.167 µM assuming trimerization. Assays were performed in triplicate at 25°C, at a flow rate of 75 µL/min and with an injection time of 5 min. The dissociation constant ($K_D$) was calculated by non-linear regression using SigmaPlot 11 software (Systat Software Inc.). Continuous fluorescence GTase assays and measurement of TPase activity using radiolabelled lipid II substrate were performed as described previously (*Bertsche et al., 2005*; *Banzhaf et al., 2012*) with slight modification. Triton X-100 concentration varied, depending on constituent proteins, from 0.04 to 0.075%. Fold-increases in GTase rates were calculated against the mean rate obtained with PBP1B alone at the same reaction conditions, at the fastest rate.

## In vitro cross-linking—pulldown and non-reversed immunoblot approaches

Proteins were mixed at appropriate concentrations in 200 µL of binding buffer (10 mM HEPES/NaOH, 10 mM $MgCl_2$, 150 mM NaCl, 0.05–0.09 % Triton X-100, pH 7.5). PBP1B or His-PBP1B at 1 µM was used, with 2 µM LpoB(sol) or His-LpoB(sol), 1 µM TolA or TolA-His and 4 µM CpoB or His-CpoB where indicated. Samples were incubated at room temperature for 10 min before addition of 0.2% wt/vol formaldehyde (Sigma-Aldrich) and further incubation at 37°C for 10 min. Excess cross-linking was blocked by addition of 100 mM Tris/HCl, pH 7.5. At this point the two approaches diverge. For the pulldown approach, samples were applied to 100 µL of washed and equilibrated Ni-NTA superflow beads (QIAGEN, Hilden, Germany) and incubated overnight at 4°C, with mixing. Beads were then washed with 6 × 1.5 mL wash buffer (10 mM HEPES/NaOH, 10 mM $MgCl_2$, 500 mM NaCl, 50 mM imidazole, 0.05% Triton X-100, pH 7.5). Retained proteins were eluted by directly boiling beads in SDS-PAGE loading buffer; beads were then removed, and samples resolved by SDS-PAGE. Gels were stained with Coomassie brilliant blue (Roth, Germany). For the non-reversed immunoblot approach, samples were resolved by non-denaturing 4–20% gradient SDS-PAGE and transferred to nitrocellulose membrane by western blot. Proteins were identified using specific antibodies; detection was with ECL prime chemiluminescence (GE Healthcare Bio-Sciences) imaged using an ImageQuant LAS4000mini (GE Healthcare Bio-Sciences).

## In vivo DTSSP cross-linking and co-immunoprecipitation assays (related to *Figure 7*)

Assays were performed as described (*Bertsche et al., 2006*; *Typas et al., 2010*) with minor modifications: Cells were grown overnight, diluted 1:400 into 250 mL Lennox LB, and grown with shaking in baffled flasks at 30°C to $OD_{600} = 0.3–0.4$. Cells were then pelleted by centrifugation and resuspended at density of $\sim 10^{10}$ cells/mL (i.e., equivalent to $OD_{600} = 10$) in 8.0-mL ice-cold CL Buffer I (50 mM $NaH_2PO_4$ 20% [wt/vol] sucrose pH 7.4) with 100 µg/mL DTSSP (freshly prepared as a 20 mg/mL stock in CL Buffer I). Cells were incubated at 4°C with mixing for 1 hr, then pelleted and frozen at −80°C. Cells were then thawed, resuspended at $OD_{600} = 4$ in ice-cold CL Buffer II (100 mM Tris–HCl 10 mM MgCl2, 1 M NaCl pH 7.5) with 100 µM PMSF, 50 µg/mL protease inhibitor cocktail (P8465, Sigma-Aldrich) and 50 µg/mL DNAse I, and lysed using a microfluidizer. Otherwise, method was as described in (*Bertsche et al., 2006*; *Typas et al., 2010*). Antibody amounts for immunoprecipitation were optimized to capture all detectible target protein, as assessed by western blot of pellet and supernatant fractions. Pelleted material was extracted in 1/10 original volume SDS-PAGE buffer to concentrate the final IP fraction.

## Data-driven structural docking models (related to *Figure 8A*)

The docking models of CpoB/PBP1B/LpoB complex were built using HADDOCK2.1 data-driven docking protocols (*Dominguez et al., 2003*) and CNS1.2 (*Brunger et al., 1998*) for the structure calculations. The initial coordinates of the CpoB, PBP1B, and LpoB molecules were taken from the PBP1B crystal structure (PDB code 3VMA) (*Sung et al., 2009*), the CpoB crystal structure from the N-terminal domain (2XDJ) and the C-terminal TPR domain (2XEV) (*Krachler et al., 2010*), and from the LpoB NMR structure (2MII) (*Egan et al., 2014*). During the multi-body docking, ambiguous restraints were applied to drive the docking according to the different experimental data. For LpoB, ambiguous interaction restraints were defined based on the NMR chemical shift perturbation mapping recorded on the LpoB/UB2H complex and on the in vivo and in vitro activities of LpoB and PBP1B alleles as described in *Egan et al. (2014)*. For CpoB, in vivo cross-linking detected using *p*Bpa substitutions at PBP1B residues 118, 123, 751, and 753 were used as active restraints to drive the interaction between PBP1B and CpoB. Due to the absence of a unique structure of the disordered N-terminal domain of LpoB (residues 1–60) and the linker between the N- and C-terminal domains of CpoB (residues 91–108), the two regions were treated as fully flexible.

The docking was performed with default HADDOCK parameters including a clustering cutoff of 7.5 Å. The HADDOCK score was used to rank the generated models (*Lutje Hulsik et al., 2013*). The generated docking models were analyzed within the Pymol software.

## Determination of CpoB copy number (related to *Figure 5—figure supplement 2*)

BW25113 and BW25113 Δ*cpoB* were grown in 50 mL of LB to an $OD_{578}$ of 0.3. The culture was then back-diluted 1:50 into 50 mL fresh LB and grown to an $OD_{578}$ of 0.3. Cells were then cooled on ice for 10 min prior to harvesting 5 mL by centrifugation ($10,000 \times g$, 10 min, 4°C). At this point, a viable count was also performed to determine the number of cells per milliliter of culture (see below). The pellets were resuspended in 100 µL TBS (Tris-buffered saline) buffer and lysed by addition of 100 µL SDS-PAGE loading buffer and boiling for 10 min. $3 \times 20$ µL samples of BW25113 lysate were resolved by SDS-PAGE along with purified CpoB standards (0, 1, 2, 4, 8, and 16 ng) loaded in 20 µL of BW25113 Δ*cpoB* lysate. CpoB was detected with specific antibody after western blot (*Figure 5—figure supplement 2*). Images were analyzed using ImageQuant LAS4000 software, giving the chemiluminescence signal derived from CpoB bands over the background (the signal at the same position as CpoB in the 0 ng sample was used as the background). A standard curve was plotted using the known CpoB standards and the amount of CpoB in the BW25113 lysate samples was calculated. This was then converted to molecules per cell using the viable counts, which were performed as follows: Cells were serially diluted 10-fold in ice-cold LB. 100 µL of the $10^{-5}$ and $10^{-6}$ dilutions were plated on LB agar and grown overnight at 37°C; colonies were then counted and the number of cells per milliliter of culture was calculated.

## Acknowledgements

We thank Adeline Derouaux and Tom Bernhardt for reagents and strains. We thank Monica Guo and Jason Peters for critical reading of the manuscript.

## Additional information

### Funding

| Funder | Grant reference | Author |
|--------|-----------------|--------|
| Wellcome Trust | 101824/Z/13/Z | Waldemar Vollmer |
| National Institutes of Health (NIH) | 5RO1GM102790 | Carol A Gross |
| National Institutes of Health (NIH) | 1P50GM107615-01 | Kerwyn Casey Huang |
| Alexander von Humboldt-Stiftung | Sofja Kovalevskaja Award | Athanasios Typas |
| European Molecular Biology Laboratory | internal funding | Athanasios Typas |
| European Commission | DIVINOCELL project FP7-Health-2007-B-223431 | Tanneke den Blaauwen |
| Nederlandse Organisatie voor Wetenschappelijk Onderzoek | Council for Chemical Sciences ECHO project # 700.59.005 | Eefjan Breukink |
| National Science Foundation (NSF) | CAREER Award MCB-1149328 | Kerwyn Casey Huang |
| Stanford University | Graduate Fellowship | Alexandre Colavin |
| Netherlands Organisation for Scientific Research | Proteins At Work, National Roadmap Large-scale Research Facilities of the Netherlands, Project number 184.032.201 | A F Maarten Altelaar |

The funders had no role in study design, data collection and interpretation, or the decision to submit the work for publication.

### Author contributions

ANG, AJFE, Conception and design, Acquisition of data, Analysis and interpretation of data, Drafting or revising the article; IL'V, AC, Acquisition of data, Analysis and interpretation of data, Drafting or revising the article; JV, AK, JB, AFMA, MJD, Acquisition of data, Analysis and interpretation of data; KCH, J-PS, Analysis and interpretation of data, Drafting or revising the article; EB, TB, AT, CAG, WV, Conception and design, Analysis and interpretation of data, Drafting or revising the article

### Author ORCIDs

Alexandra Koumoutsi, http://orcid.org/0000-0001-8368-4193

## Additional files

### Supplementary file

• Supplementary File 1. Bacterial strains and plasmids used in this study.

### Major datasets

The following previously published datasets were used:

| Author(s) | Year | Dataset title | Dataset ID and/or URL | Database, license, and accessibility information |
|-----------|------|---------------|------------------------|--------------------------------------------------|
| Sung MT, Lai YT, Huang CY, Chou LY, Shih HW, Cheng WC, Wong CH, Ma C | 2009 | Crystal Structure of the Full-Length Transglycosylase PBP1b from Escherichia coli | http://www.rcsb.org/pdb/explore/explore.do?structureId=3VMA | Publicly available at RCSB Protein Data Bank (3VMA). |

| Author(s) | Year | Dataset title | Dataset ID and/or URL | Database, license, and accessibility information |
|---|---|---|---|---|
| Krachler AM, Sharma A, Cauldwell A, Papadakos G, Kleanthous C | 2010 | Crystal structure of the N-terminal domain of E. coli YbgF | http://www.rcsb.org/pdb/explore/explore.do?structureId=2XDJ | Publicly available at RCSB Protein Data Bank (2XDJ). |
| Krachler AM, Sharma A, Cauldwell A, Papadakos G, Kleanthous C | 2010 | Crystal structure of the TPR domain of Xanthomonas campestris YbgF | http://www.rcsb.org/pdb/explore/explore.do?structureId=2XEV | Publicly available at RCSB Protein Data Bank (2XEV). |
| Egan AJ, Jean NL, Koumoutsi A, Bougault CM, Biboy J, Sassine J, Solovyova AS, Breukink E, Typas A, Vollmer W, Simorre JP | 2014 | NMR structure of E. coli LpoB | http://www.rcsb.org/pdb/explore/explore.do?structureId=2MII | Publicly available at RCSB Protein Data Bank (2MII). |

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
