## [Decision Letter]

Thank you for sending your work entitled “Coordination of peptidoglycan synthesis and outer membrane constriction during *Escherichia coli* cell division” for consideration at *eLife*. Your article has been favorably evaluated by Richard Losick (Senior editor), a Reviewing editor, and three reviewers. The reviewers discussed their comments before we reached this decision, and the Reviewing editor has assembled the following comments to help you prepare a revised submission.

Overall, this is an excellent and very thorough study that provides important and novel insights into the biogenesis of the gram-negative cell envelope during cell division. The findings break new ground in the fundamental understanding of bacterial cell division. Also, given the role of the gram-negative cell envelope in antibiotic resistance, the results are also of practical significance. However, there are some areas where the interpretation of the data needs to be more balanced, where the manuscript should be simplified and some additional controls/information need to be included.

1) The weak phenotype associated with a CpoB deficiency needs to be discussed explicitly. Only a modest elevation in cefsulodin sensitivity and a quite low frequency of cells (1% or less) with membrane blebs are observed. Sensitivity to many drugs would be more apparent if the envelope defect was strong. The relatively weak phenotype does not diminish the findings in this report. It only means that other partially redundant factors (possibly like LpoA) likely remain to be characterized.

2) The description of the crosslinking data in Figure 5 (presented in the figure with the limited range in the y-axes) also needs to be more balanced. The observed changes may be statistically significant, but the text suggests there is a much more dramatic effect than is actually observed. (For example, in the last paragraph of the subsection headed “CpoB and TolA modulate PBP1B-mediated PG synthesis” the authors state: “…under this condition [215 mM NaCl], CpoB completely prevented PBP1b from synthesizing hyper-crosslinked PG, and TolA again completely alleviated this effect…”.) The total change is only 10%, from 60% crosslinking, down to 50% with CpoB, and back again with TolA addition. The authors need to explicitly state that the observed changes are modest and provide some discussion to indicate that, even so, the trends in the activity suggest an interesting underlying mechanism that quite possibly may have a more significant effect on crosslinking in vivo.

3) The text related to ternary and quaternary complex formation should similarly be moderated given that the data in Figure 5—figure supplement 1 do not make a compelling case for these conclusions. In Figure 5—figure supplement 1, the authors present the data to support their conclusion that CpoB or TolA can only interact with LpoB in the presence of PBP1b. In the top panel, a vanishingly small amount of CpoB co-purifies with His-LpoB in the presence of PBP1b. No CpoB co-purifies with His-LpoB without PBP1b, but the yield of His-LpoB is much lower in this sample. Given the low levels of CpoB that co-purified initially, it is hard to know if the lack of interaction here is due to a yield problem. Also, the CpoB that does co-purify in the presence of PBP1b does not look like it is much over background and could just be a small amount of spurious crosslinking. The same is true for the TolA and TolA+CpoB experiments. For Figure 5—figure supplement 1, it is difficult to interpret the composition of the crosslinked complexes without the inclusion of pairwise samples (i.e. PBP1b + CpoB with and without TolA etc.).

4) The data in the paper show a connection between the energy transducing TolAQR complex and PBP1b mediated by CpoB, and this is important advance. However, as evidenced by the title, the authors go beyond this to claim that CpoB coordinates peptidoglycan synthesis with OM constriction. Indeed, throughout the paper the phrase Tol-Pal is used repeatedly, and as shown in Figure 7, Pal is the critical OM protein that would be responsible for this constriction. The fact is that no experiments with Pal (or with TolB, Figure 7) are presented in the entire paper. The OM constriction claim is based on the common assumption that TolA interacts directly with the OM lipoprotein Pal or indirectly with Pal through the periplasmic protein TolB, as implied by the Tol-Pal phrase and as shown in Figure 7, and this assumption may be correct. However, this assumption has been challenged ([9] EMBO J. 28:2846). Bonsor et al. could detect no interaction between TolA and Pal, and Pal was shown to inhibit the interaction between TolA and TolB, and these results question the existence of a transenvelope Tol-Pal complex. Accordingly, the authors need to be more cautious about the claim made in the title, the renaming of YbgF, the model shown in Figure 7, and the use of the Tol-Pal phrase.

5) The first part of the paper describing the synthetic lethality of lpoA cpoB mutations is confusing. The authors should simply state the phenotypes of lpoA and mrcA mutants in combination with cpoB are not equivalent, and provide a short discussion of this and what it might mean.

6) Figure 4: A specificity control using PBP1a should be included here to rule out the possibility that the observation is a result of TolA and PBP1b in the same detergent micell. Thus, when TolA has no TM domain, it doesn't co-purify.

7) Figure 6: More information about the extract samples is needed. Do they represent the input for the immunoprecipitations? Are the intensities directly comparable to the intensities in the IP samples? It would be helpful to know what percentage of the input protein actually ends up being IP-ed. Is it a significant fraction? Or a very small one?

8) All of the in vitro experiments with full-length TolA have been performed in the absence of significant concentrations of detergent. Is it established that TolA does not form aggregates or interact non-specifically through its hydrophobic transmembrane helix under these conditions?

Reviewer #1:

1) I might have missed this, but I don't think it was shown that cpoB in trans corrects the membrane and/or other defects of the deletion mutant. Given that the gene is present in the *tol-pal* locus, it is probably important to show the complementation to rule out the possibility that the deletion adversely affects *tol-pal* expression.

Reviewer #2:

This paper, together with the supplemental data, is very long, very dense, and the model shown in Figure 7 contains 8 different proteins located in 3 different cellular compartments. Moreover, for 2 of the proteins the gene name and protein name don't coincide. To make it understandable for the general reader of *eLife*, a cartoon depicting these proteins and their cellular location is needed very early in the presentation. The work showing the second function of LpoA is a beautiful story. However, it adds another layer of complexity and I'm not sure that it is critical for the story presented.

Reviewer #3:

1) How have the authors chosen the residues for photo-crosslinking in Figure 2? It would be helpful for the reader to get some information on the reasoning behind this experiment.

2) Why are the signals for the deletion mutants in Figure 4—figure supplement 5 so different from those of the wild-type strain?

3) In the SPR experiment (Figure 4), not all of the protein dissociates from the chip after washing with buffer. I wonder whether it is correct to simply use the plateau levels of the different curves to calculate the K_D_ value if the total peak level apparently reflects more than one distinct complex? This is not really relevant for the conclusion that PBP1B and CpoB interact, but if a K_D_ value is given, it should be made sure that it is correct.

4) In case TolA(sol) does indeed not interact with PBP1B, using this protein would be a good control for the specificity of the effect on PBP1B activity observed with full-length TolA (Figure 5).

5) I wonder how reliable the kinetic rates in Figure 6—figure supplement 1 are. They appear to vary significantly between the different graphs. Moreover, the values given appear to be based on the bottom graph in which there was apparently some disturbance during the first 400 sec of the measurement.

6) The authors use the results from their in vivo cross-linking experiments to conclude on direct interactions between proteins (in the subsection headed “Regulation of PBP1B by CpoB and TolA responds to Tol-Pal energy state in vivo”). In general, in this kind of analysis, it is difficult to exclude the possibility that these interactions are indirect, in particular when using a relatively long cross-linker such as DTSSP.

---

## [Author Response]

*1) The weak phenotype associated with a CpoB deficiency needs to be discussed explicitly. Only a modest elevation in cefsulodin sensitivity and a quite low frequency of cells (1% or less) with membrane blebs are observed. Sensitivity to many drugs would be more apparent if the envelope defect was strong. The relatively weak phenotype does not diminish the findings in this report. It only means that other partially redundant factors (possibly like LpoA) likely remain to be characterized*.

The reviewers expressed concern about the relatively weak phenotypes of a *cpoB* deletion, and we now address this explicitly in the Discussion (in the subsection headed “Why is PBP1B tuning necessary for coordinating septal PG synthesis and OM constriction?”). We agree that this is likely due to redundant factors, in particular LpoA. It is also worth noting, however, that Δ*cpoB* is sensitive to a number of other drugs and conditions, including almost all β-lactams to which Δ*mrcB* and Δ*lpoB* are sensitive (Nichols et al., 2010 and Figure 2); these multiple shared sensitivities drive the strong correlation between phenotype profiles of Δ*cpoB* and Δ*mrcB* strains. Also of note, the lysis phenotype of Δ*cpoB* is greatly exacerbated under low-salt conditions (Figure 2—figure supplement 1). Thus, while Δ*cpoB* single-mutant phenotypes are relatively mild under ideal lab conditions, in the normal environment of *E. coli*, osmotic stress and other perturbations are likely to exacerbate these phenotypes. We also now make these points more explicitly.

*2) The description of the crosslinking data in*
Figure 5
*(presented in the* figure *with the limited range in the y-axes) also needs to be more balanced. The observed changes may be statistically significant, but the text suggests there is a much more dramatic effect than is actually observed. (For example, in the last paragraph of the subsection headed “CpoB and TolA modulate PBP*1B*-mediated PG synthesis” the authors state: “…under this condition [215 mM NaCl], CpoB completely prevented PBP*1b *from synthesizing hyper-crosslinked PG, and TolA again completely alleviated this effect…”.) The total change is only 10%, from 60% crosslinking, down to 50% with CpoB, and back again with TolA addition. The authors need to explicitly state that the observed changes are modest and provide some discussion to indicate that, even so, the trends in the activity suggest an interesting underlying mechanism that quite possibly may have a more significant effect on crosslinking in vivo*.

We have changed the manuscript to address this comment (subsection “CpoB and TolA modulate PBP1B-mediated PG synthesis”): “…In this condition, LpoB increased cross-linking from 50 to 60%. The addition of CpoB completely prevented this stimulation, whereas TolA again alleviated the CpoB effect (Figure 6, right graph).”

Although the magnitudes of the effects in the in vitro TPase assay may seem moderate, it is helpful to keep in mind that these are end-point effects, and do not preclude additional rate effects. In addition, these effects are likely significant in vivo: LpoB has only a 10% effect in vitro in these conditions, but is essential for PBP1B function in vivo, under all salt conditions we tested ([84]; also [68]). Similarly, all Δ*cpoB* in vivo phenotypes are rescued by overexpressing PBP1B (Figure 2). We have made revisions to make these points more clear and explicit in the text (in the aforementioned subsection).

*3) The text related to ternary and quaternary complex formation should similarly be moderated given that the data in*
Figure 5—figure supplement 1
*do not make a compelling case for these conclusions. In*
Figure 5—figure supplement 1*, the authors present the data to support their conclusion that CpoB or TolA can only interact with LpoB in the presence of PBP*1b*. In the top panel, a vanishingly small amount of CpoB co-purifies with His-LpoB in the presence of PBP*1b*. No CpoB co-purifies with His-LpoB without PBP*1b*, but the yield of His-LpoB is much lower in this sample. Given the low levels of CpoB that co-purified initially, it is hard to know if the lack of interaction here is due to a yield problem. Also, the CpoB that does co-purify in the presence of PBP*1b *does not look like it is much over background and could just be a small amount of spurious crosslinking. The same is true for the TolA and TolA+CpoB experiments. For*
Figure 5—figure supplement 1*, it is difficult to interpret the composition of the crosslinked complexes without the inclusion of pairwise samples (i.e. PBP*1b *+ CpoB with and without TolA etc.)*.

Indeed, in the top panel the yield of His-LpoB was low, and even lower in the His- LpoB/CpoB sample, presumably due to extensive washing of the Ni-beads with buffer containing 50 mM imidazole. We therefore repeated the experiment twice, with less washing, obtaining similar yields of bound His-LpoB in the His-LpoB/PBP1B/CpoB and His-LpoB/CpoB samples. The level of CpoB retained in the His-LpoB/PBP1B/CpoB is clearly above background. The middle panel (with TolA) shows with the His-LpoB sample all untagged proteins clearly above background. In the lower panel, we agree that the experiment with all four proteins together does not prove the simultaneous binding of CpoB, LpoB and TolA to PBP1B to form a quarternary complex, and we have therefore removed this gel to focus on the three ternary complexes, which are supported by the data shown in Figure 5—figure supplement 1. This does not require change of the text (please see the subsection entitled “CpoB, TolA, PBP1B, and LpoB form a complex”).

*4) The data in the paper show a connection between the energy transducing TolAQR complex and PBP*1b *mediated by CpoB, and this is important advance. However, as evidenced by the title, the authors go beyond this to claim that CpoB coordinates peptidoglycan synthesis with OM constriction. Indeed, throughout the paper the phrase Tol-Pal is used repeatedly, and as shown in*
Figure 7*, Pal is the critical OM protein that would be responsible for this constriction. The fact is that no experiments with Pal (or with TolB,*
Figure 7*) are presented in the entire paper. The OM constriction claim is based on the common assumption that TolA interacts directly with the OM lipoprotein Pal or indirectly with Pal through the periplasmic protein TolB, as implied by the Tol-Pal phrase and as shown in*
Figure 7*, and this assumption may be correct. However, this assumption has been challenged (*[9]
*EMBO J. 28:2846). Bonsor et al. could detect no interaction between TolA and Pal, and Pal was shown to inhibit the interaction between TolA and TolB, and these results question the existence of a transenvelope Tol-Pal complex. Accordingly, the authors need to be more cautious about the claim made in the title, the renaming of YbgF, the model shown in*
Figure 7*, and the use of the Tol-Pal phrase*.

We agree that there is controversy in the literature about Tol-Pal interactions (e.g., Cascales et al*.*, 2000 vs. Bonsor et al*.*, 2009) and thank the reviewers for raising this important issue. While we feel it is clear that TolQRA/Pal localize to mid-cell during division and contribute to OM constriction ([36], and Figure 4, TolA), we agree that how they do so is not yet clear at a mechanistic level, and we now acknowledge this explicitly in the text and cite [9] (Introduction). We now avoid assumptions about trans-envelope Tol-Pal interactions in the text, and refer to the ‘Tol apparatus’ or ‘TolQRA complex’ rather than ‘Tol-Pal’ where appropriate. We also now mention that Tol and CpoB may have additional/other roles during PG synthesis (please see the subsection headed “Why is PBP1B tuning necessary for coordinating septal PG synthesis and OM constriction?”). We do prefer, however, to keep the name ‘CpoB’ because: 1) CpoB localizes to mid-cell during OM constriction, concurrent with TolA and PBP1B; 2) OM defects we observe are also consistent with a role in OM constriction; 3) the name does not assume a specific mechanism of Tol function, only a role in OM constriction; and 4) overall, we feel the name encapsulates our findings and the functional role we have identified.

*5) The first part of the paper describing the synthetic lethality of lpoA cpoB mutations is confusing. The authors should simply state the phenotypes of lpoA and mrcA mutants in combination with cpoB are not equivalent, and provide a short discussion of this and what it might mean*.

We thank the reviewers for pointing this out, and have now recast and simplified this portion of the Results by moving the double mutant analysis and its implications to a new section entitled “Genetic interactions further implicate CpoB in PBP1B function and reveal a CpoB-related function for LpoA”.

*6)*
Figure 4*: A specificity control using PBP*1a *should be included here to rule out the possibility that the observation is a result of TolA and PBP*1b *in the same detergent micell. Thus, when TolA has no TM domain, it doesn't co-purify*.

This is a good point, and we thank the reviewers for this comment. We have adjusted the text (subsection “CpoB, TolA, PBP1B, and LpoB form a complex”) to not overstate the observation that the TM domain is required for binding: “…suggesting that domain I of TolA is important for the interaction.”

With regard to ensuring specificity using His-PBP1A, Figure 5—figure supplement 1 features a sample of PBP1A incubated with TolA (and CpoB) at the same conditions as for Ni-NTA bead pulldown experiments, and we see no significant cross-linking. We also argue that the in vivo co-immunoprecipitation of PBP1B and TolA supports a direct interaction between the two proteins, given that the interaction is still seen in the absence of CpoB (Figure 6).

Further, to be more sure of specificity, we have calculated the ratio of protein to detergent micelles and performed a series of controls to ensure that the cross-linking is specific, and not due to a limited number of available detergent micelles. We used the information in: Robson and Dennis. 1977. The size, shape, and hydration of nonionic surfactant micelles. Triton X-100. *The Journal of Physical Chemistry* 81:1075–1078.

A sample containing 0.1% Triton X-100 has ∼11.1 µM micelles: micelle/protein ratio of ∼5:1. The assay shown in Figure 6 was performed at 0.09% Triton X-100, a fact which we now note properly in the Methods. We still see the interaction when we shift the ratio, increasing the number of empty micelles, to the extreme at 2% Triton (221 µM micelles, micelle/protein ratio of ∼110:1) (see Figure 9).

Author response image 1.**DOI:**
http://dx.doi.org/10.7554/eLife.07118.027

*7)*
Figure 6*: More information about the extract samples is needed. Do they represent the input for the immunoprecipitations? Are the intensities directly comparable to the intensities in the IP samples? It would be helpful to know what percentage of the input protein actually ends up being IP-ed. Is it a significant fraction? Or a very small one?*

The extracts are the Triton-soluble membrane fractions, which were used as input for the IPs. We have adjusted the figure and legend to make this clear. Because the IP fractions are concentrated (to deal with the low DTSSP crosslinking), they are not directly comparable to the extracts. The IPs are, however, directly comparable across mutants, as all were concentrated to the same degree. Additionally, we optimized the experiments such that all detectable target protein (as determined by western blot) was immunoprecipitated. We have added additional details to the figure legend and Methods (in the subsection headed “In vivo DTSSP cross-linking and co-immunoprecipitation assays”) to make this clear as well.

*8) All of the in vitro experiments with full-length TolA have been performed in the absence of significant concentrations of detergent*. *Is it established that TolA does not form aggregates or interact non-specifically through its hydrophobic transmembrane helix under these conditions?*

All experiments with full-length TolA were done in the presence of 0.06 - 0.075% Triton X-100, which is significantly above the CMC (0.02%) and in the range of what is often used to prevent non-specific interactions (we now note Triton X-100 concentrations properly in the Methods, in the subsections “In vitro protein interaction and activity assays” and “In vitro cross-linking—pulldown and non-reversed immunoblot approaches”). TolA-his or TolA at the conditions used in the in vitro assays were centrifuged at 350,000 × *g* for 30 min at 28°C. Supernatant samples were taken before and after centrifugation, both were resolved by SDS-PAGE and stained with Coomassie blue. There was no change in the amount of TolA in the supernatant, suggesting that big aggregates are absent (see Figure 10).

Author response image 2.**DOI:**
http://dx.doi.org/10.7554/eLife.07118.028

Regarding the specificity of the effect on enzyme activity, we have performed in vitro activity assays in which PBP1B (± LpoB) was incubated with PBP3 (possessing it's TM helix) at 0.06% Triton and we see no effect.

Reviewer #1:

*1) I might have missed this, but I don't think it was shown that cpoB in trans corrects the membrane and/or other defects of the deletion mutant. Given that the gene is present in the* tol-pal *locus, it is probably important to show the complementation to rule out the possibility that the deletion adversely affects* tol-pal *expression*.

We have added *cpoB* complementation data (new Figure 2—figure supplement 2).

Reviewer #2:

*This paper, together with the supplemental data, is very long, very dense, and the model shown in*
Figure 7
*contains 8 different proteins located in 3 different cellular compartments. Moreover, for 2 of the proteins the gene name and protein name don't coincide. To make it understandable for the general reader of* eLife*, a cartoon depicting these proteins and their cellular location is needed very early in the presentation. The work showing the second function of LpoA is a beautiful story. However, it adds another layer of complexity and I'm not sure that it is critical for the story presented*.

We thank the reviewer for this feedback, and have added an early figure illustrating the involved proteins (new Figure 1). Regarding the second function of LpoA, we have simplified this portion of the text (see Major Point 5). We feel it is important to include this data, however, in particular because it explains why Δ*lpoA* Δ*ybgF* and Δ*mrcA* Δ*ybgF* exhibit different phenotypes.

Reviewer #3:

*1) How have the authors chosen the residues for photo-crosslinking in*
Figure 2*? It would be helpful for the reader to get some information on the reasoning behind this experiment.*

We now comment on this in the text (in the beginning of the Results section). We initially modified 18 surface exposed residues of the UB2H domain, identifying 1 (118) cross-linking with CpoB. Next, we modified 6 further residues near T118; of these, E123 cross-linked with CpoB. Then we mutated opposing amino acids (with respect to E123 and T118) on the TP domain (T751 and T753) and found that they also cross-linked with CpoB.

*2) Why are the signals for the deletion mutants in*
Figure 4—figure supplement 5
*so different from those of the wild-type strain?*

In Δ*tolA*, CpoB levels are in fact elevated (∼25% over wild type), and when TolA is overexpressed they are reduced. (We have observed this both by immunofluorescence, as shown here, and by western blot.) The images that were used to assemble the Δ*tolA* profile map were taken using a shorter exposure time in part to compensate for this, which is why that image profile appears less bright. We note this now in the figure legend. We have also added a plot of fluorescence vs. normalized cell length to clarify this difference; the quantification in this plot accounts for the difference in exposure times. We do not know why TolA has this effect on CpoB levels, though we would speculate that it is an effect of TolA on CpoB stability that is tied to its effect on CpoB oligomeric state (48).

*3) In the SPR experiment (*Figure 4*), not all of the protein dissociates from the chip after washing with buffer. I wonder whether it is correct to simply use the plateau levels of the different curves to calculate the K*_*D*_
*value if the total peak level apparently reflects more than one distinct complex? This is not really relevant for the conclusion that PBP*1B *and CpoB interact, but if a K*_*D*_
*value is given, it should be made sure that it is correct*.

We thank the reviewer for this comment. Indeed, we do not know the stoichiometry of the PBP1B-CpoB interaction. PBP1B can dimerize, and CpoB is a trimer but forms a 1:1 complex with TolA (48). In our SPR experiment CpoB could bind as monomer or trimer to PBP1B (which is possibly a monomer on the chip surface because of the high NaCl concentration during immobilization and the low surface load), and a possible complex of PBP1B with a CpoB trimer could lose CpoB molecules at different rates during the dissociation phase. We do not observe a decline in the RU signal already during the association phase excluding the possibility of a fast binding of a CpoB trimer to PBP1B followed by slow dissociation of one or two CpoB molecules. Taking into account these points, we have now re-analysed the SPR data and calculated the K_D_ with the concentration of the CpoB trimer (which is the most likely scenario), and changed the manuscript (in the subsection headed “CpoB, TolA, PBP1B, and LpoB form a complex” and Figure 5) to indicate that we present the *apparent* K_D_ of the interaction of PBP1B with a CpoB trimer.

*4) In case TolA(sol) does indeed not interact with PBP*1B*, using this protein would be a good control for the specificity of the effect on PBP*1B *activity observed with full-length TolA (*Figure 5*)*.

We have now done this control showing that, indeed, TolA(sol) has no effect on PBP1B activity (see Figure 11). We also performed a reaction with a large excess of TolA(sol) and CpoB together (both at 40 µM) and saw no effect (not shown).

Author response image 3.*>* Contineous GTase assay with dansyl-lipid II and 1 µM PBP1B with 3 µM TolA-his or TolA(sol) at 0.065% Triton, 30°C.**DOI:**
http://dx.doi.org/10.7554/eLife.07118.029

*5) I wonder how reliable the kinetic rates in*
Figure 6—figure supplement 1
*are. They appear to vary significantly between the different graphs. Moreover, the values given appear to be based on the bottom graph in which there was apparently some disturbance during the first 400 sec of the measurement*.

We mention in the legend that the variation is due to different conditions (PBP1B concentration, temperature, detergent concentration) optimized for each experiment. For example, with LpoB/TolA the reaction is too fast at the standard conditions to resolve the initial reaction rate; therefore, in this experiment the PBP1B concentration and temperature were lower to allow for rate determination. We did not use the bottom graph for all calculations. For Figure 6 the fold-increases in rates were always calculated against the mean rate obtained with PBP1B alone at the same reaction conditions, at the fastest rate (which was after 400 sec in the lower graph of Figure 6—figure supplement 1). We have now clarified this in the Methods (in the subsection headed “In vitro protein interaction and activity assays”).

*6) The authors use the results from their in vivo cross-linking experiments to conclude on direct interactions between proteins (in the subsection headed “Regulation of PBP*1B *by CpoB and TolA responds to Tol-Pal energy state in vivo”). In general, in this kind of analysis, it is difficult to exclude the possibility that these interactions are indirect, in particular when using a relatively long cross-linker such as DTSSP*.

Indeed, we cannot formally exclude indirect in vivo cross-linking with DTSSP. However, the low cross-linking activity and the fact that we observe the interactions with purified proteins and in the absence of possible bridging partners suggest that these are direct interactions. Therefore, we did not change the text of the revised manuscript.